# Neural activity in a hippocampus-like region of the teleost pallium is associated with active sensing and navigation

Haleh Fotowat[1]*, Candice Lee[2], James Jaeyoon Jun[3], Len Maler[2]

[1]Department of Molecular and Cellular Biology, Harvard University, Cambridge, United States; [2]Department of Cellular and Molecular Medicine, Brain and Mind Institute and Centre for Neural Dynamics, University of Ottawa, Ottawa, Canada; [3]Center for Computational Biology, Flatiron Institute, New York, United States

**Abstract** Most vertebrates use active sensing strategies for perception, cognition and control of motor activity. These strategies include directed body/sensor movements or increases in discrete sensory sampling events. The weakly electric fish, *Gymnotus sp.*, uses its active electric sense during navigation in the dark. Electric organ discharge rate undergoes transient increases during navigation to increase electrosensory sampling. *Gymnotus* also use stereotyped backward swimming as an important form of active sensing that brings objects toward the electroreceptor dense fovea-like head region. We wirelessly recorded neural activity from the pallium of freely swimming *Gymnotus*. Spiking activity was sparse and occurred only during swimming. Notably, most units tended to fire during backward swims and their activity was on average coupled to increases in sensory sampling. Our results provide the first characterization of neural activity in a hippocampal (CA3)-like region of a teleost fish brain and connects it to active sensing of spatial environmental features.

DOI: https://doi.org/10.7554/eLife.44119.001

*For correspondence:
halehfotowat@fas.harvard.edu

**Competing interests:** The authors declare that no competing interests exist.

## Introduction

Many animals, for example rodents, actively explore their environment in order to learn its spatial structure (*Alvernhe et al., 2012*; *Gordon et al., 2014*; *Saraf-Sinik et al., 2015*; *Amey-Özel et al., 2015*). Several forms of active sensing by rodents, including eye and head scanning motions and whisking, are used during navigation (*Moore et al., 2013*; *Lever et al., 2006*; *Poulter et al., 2018*) and may be indicators of attentional processes linked to spatial learning (*Aly and Turk-Browne, 2016*; *Muzzio et al., 2009*). The hippocampus is essential for spatial learning and memory (*Barry and Burgess, 2014*; *Poulter et al., 2018*; *O'Keefe and Nadel, 1978*). Although attention has been linked to memory formation, storage and retrieval by the hippocampus (*Aly and Turk-Browne, 2016*; *Muzzio et al., 2009*), there have been few attempts to link active sensing to the hippocampal manifestations of spatial learning, for example place fields and other forms of spatially specific discharge by hippocampal/entorhinal neurons (see Discussion).

Teleost fish, whose common ancestor with mammals lived approximately 450 million years ago are also capable of spatial learning (*Rodríguez et al., 2002*; *Jun et al., 2016*). Whether these fish utilize the same neural mechanisms for spatial learning as mammals do remains completely unknown. In goldfish, studies based on lesion and cytochrome oxidase histochemistry have identified the dorsolateral pallium (DL) as selectively essential for spatial learning (*Ocaña et al., 2017*; *Uceda et al., 2015*; *Rodríguez et al., 2002*; *Durán et al., 2010*; *Broglio et al., 2010*). We chose to study neural activity in the pallium of the pulse-type weakly electric fish *Gymnotus sp.* in the context of spatial navigation. These fish emit discrete electric organ discharges (~1–2 ms, EOD); the EOD generates

an electric field around the fish that stimulates an array of cutaneous electroreceptors. Conductive (e.g. plants, prey) and non-conductive (e.g. rocks) objects cause local distortions of the electric field that modulate the activity of the electroreceptors. Low level electrosensory structures in the brain can extract information on the identity, location and motion of the such objects (*Clarke et al., 2015*).

In a detailed analysis of *Gymnotus* spatial learning, *Jun et al. (2016)* have shown that these fish can learn the location of food relative to landmarks in complete darkness relying mainly on their short- range (<3 cm active electrosensory system, see *Jun et al., 2016* Figure 5C). After learning was complete, the fish were able to rapidly navigate from a landmark to the remembered food site (probe trial: no food present) clearly demonstrating that they had learned the relative location of landmarks and food.

This study further demonstrated that, in the process of learning, *Gymnotus* use several active sensing strategies. *Gymnotus* can increase the EOD rate near boundaries and landmarks resulting in an increased rate of discrete sensory sampling. They can also perform stereotyped back and forth swimming movements (B-scans) past landmarks (*Jun et al., 2016*). The backward phase of B-scans brought the landmarks over the head of the fish (*Figure 1*). The highest density and large number of electroreceptors are located on the head (*Castelló et al., 2000*), and the electric field is funneled to the perioral region; this has led Caputi and colleagues (*Aguilera and Caputi, 2003*; *Sanguinetti-Scheck et al., 2011*, *Pedraja et al., 2018*) to describe the head and perioral region as an electrosensory fovea. B-scans are therefore a critical active sensing motion for *Gymnotus* likely important for spatial learning and perhaps functionally analogous to the head scanning and whisking motions of rodents. *Gymnotus* therefore provides an excellent model system for reading out the dynamics of sensory sampling and 'attentive state' and relating this information to neuronal activity associated with learning the location of landmarks during spatial navigation. All forms of active sensing near landmarks are more frequent during early trials and decrease as the fish learns the relative location of landmarks and food. This led Jun et al to hypothesize that active sensing by *Gymnotus* serves as an attentional mechanism engaged during spatial learning. The experiments described below were carried out in naive fish, that is in fish that were learning about a new environment. These results are therefore best comparable to results from the early learning stage of Jun et al. This choice was motivated by our desire to maximize observations of active sensing as the fish encountered, and presumably learned about, the location of landmarks in the experimental tank.

Changes in electric field caused by motion past landmarks are sensed by thousands of electroreceptors located on the fish's skin (*Figure 1*). Primary afferents relay this information to the electrosensory lobe (ELL) which then projects to the mid-brain torus semicircularis (TS) that then provides electrosensory input to the optic tectum (*Krahe and Maler, 2014*; *Carr et al., 1982*). Cells in the fish's tectum respond to both electrosensory and visual directional object motions (*Bastian, 1982*; *Heiligenberg and Rose, 1987*) and would therefore be expected to discharge as the fish swims past landmarks. The tectal cells responsive to object motion project to the preglomerular nucleus (PG), an analog or homolog of the mammalian thalamus (*Giassi et al., 2012b*; *Wallach et al., 2018*). Numerous PG neurons respond to the initiation of motion or continuous object motion; PG cells responsive to either visual or electrosensory motion were described by Wallach et al but only electrosensory motion responses are expected to drive PG neurons in our experiments. PG in turn projects to DL. Based on its connectivity and gene expression, DL has characteristics of both mammalian cortex and the hippocampus (*Elliott et al., 2017*). DL can be considered as a pallial input region similar to layer 4 of mammalian cortex (see *Figure 1* and *Elliott et al., 2017*). This interpretation of the circuitry leads to the central pallium (DC) being considered as a cortical output layer (L5/6) consistent with its expression of two genes found in L5/6 pyramidal cells (*Harvey-Girard et al., 2012*). DC in turn projects massively to the tectum (*Giassi et al., 2012b*) and this appears to be the only route (in gymnotiform fish) by which memories of landmark location can control locomotion from a landmark to a remembered food site (tectal motor system references in *Figure 1* Legend).

DL can also be considered as similar or homologous to the dentate gyrus (granule cells, *Elliott et al., 2017*), a view consistent with its role in forming and storing spatial memories. DL projects massively and in a highly convergent manner to the intermediate subdivision of dorsal pallium (DDi); DDi then provides strong feedback to DL via a magnocellular component of DD (DDmg, *Giassi et al., 2012c*; *Trinh et al., 2016*; *Elliott et al., 2017*). We have recently suggested that DDi has similar connectivity, and might be homologous to, the mammalian CA3 region, while DDmg is

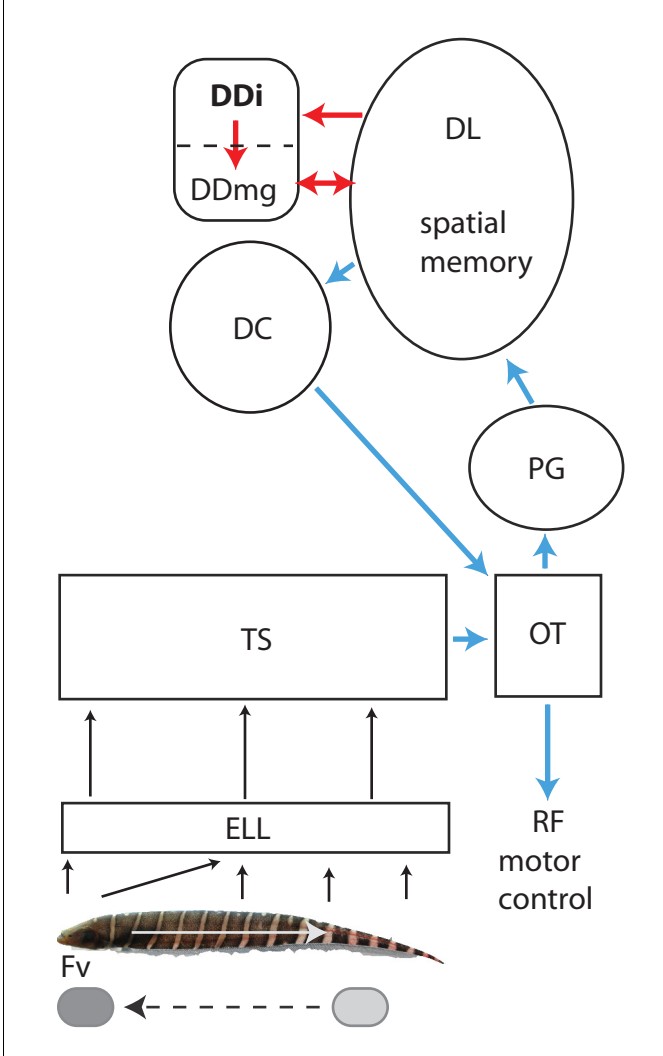

**Figure 1.** Schematic diagram of the connectivity of DDi and its relation to the neural circuitry for electrolocation. Electroreceptors are distributed over the body of a *Gymnotus sp.* with the highest density and highest number on the head; in addition, the electric currents generated by the electric organ are 'funnelled' toward the head. The 'nose' and head have therefore been considered to be the electrosensory equivalent of a fovea (*Castelló et al., 2000*; *Aguilera and Caputi, 2003*). Backward scans by the fish (white arrow) lead to relative forward motion of stationary objects (dashed black arrow) that brings them to the foveal region. Electroreceptors project topographically to the electrosensory lobe (ELL) where the foveal region is mapped to over half the ELL (*Carr et al., 1982*). The ELL projects topographically to the torus semicircularis (TS). Neurons in the ELL respond to both electrolocation and electrocommunication signals while neurons in TS are selective for one or the other of these signal categories (*Vonderschen and Chacron, 2011*). The diagram here illustrates the electrolocation pathways and ignores the electrocommunication pathways emanating from the TS (*Giassi et al., 2012b*). Blue: Neural circuitry similar to the cortical/superior colliculus circuitry of amniotes (see *Giassi et al., 2012b*; *Giassi et al., 2012c*; *Harvey-Girard et al., 2012*; *Wallach et al., 2018*; *Trinh et al., 2016*; *Elliott et al., 2017*). TS projects topographically to the tectum where cells are highly responsive to object motion (electrosensory and visual). Tectum then projects non-topographically to the preglomerular nucleus (PG, putative thalamus homolog) where a variety of motion-sensitive neurons (electrosensory and visual) have been identified (*Wallach et al., 2018*). PG projects in turn to the dorsolateral pallium (DL). DL projects to central pallium (DC) whose cells are comparable to the layer 5/6 pyramidal cells of cortex. DC then projects to tectum. Numerous tectal neurons of goldfish and zebrafish project to the reticular formation and can drive directed movements via this projection (*Herrero et al., 1998*; *Luque et al., 2008*; *Gahtan et al., 2005*).Red: Neural circuitry similar to the mammalian circuitry required for the formation of spatial maps (see *Elliott et al., 2017* for a detailed discussion). DL is essential for spatial learning and memory in goldfish and, in this view, is similar to the mammalian dentate

*Figure 1 continued on next page*

*Figure 1 continued*

gyrus. DL projects to DDi (similar to CA3). Both DL and DDi project to the large cells of DDmg (similar to dentate hilus mossy cells). DDmg then projects back to DL much like mossy cells project back onto the dentate gyrus granule cells. We have omitted, for simplicity, the GABAergic interneurons that appear to correspond to the somatostatin positive hilar interneurons of the mammalian hippocampus. We would like to thank Will Crampton for providing the image of a *Gymnotus* fish used in this figure.

DOI: https://doi.org/10.7554/eLife.44119.002

likewise comparable to the mossy cells found in the hilus of the dentate gyrus (*Elliott et al., 2017*). DL, DDi and DDmg would, from this perspective, likely contain neurons responsive to the presence of landmarks. DL has vastly more neurons than either DDi or DDmg (*Figure 2—figure supplement 1*, *Trinh et al., 2016*), and they are far more densely packed (*Giassi et al., 2012c*); DDmg is a relatively small region (*Giassi et al., 2012a*) and, in preliminary experiments, we found it difficult to selectively target it. We therefore decided that recording from DDi would be the most efficient for determining what spatial information that might be extracted by the teleost pallium via active sensing.

We used a wireless transmitter system to record neural activity from DDi as the fish swam freely in the dark in an experimental tank (open maze) containing differently shaped landmarks. Recordings were done as the fish were learning about a novel environment using the same tank and landmarks as in the *Jun et al. (2016)* study so that we could directly compare behavioral and physiological data. We simultaneously tracked the fish's position and recorded its EOD signal using electrodes placed inside the tank (*Jun et al., 2014a*). We asked: what are the neural dynamics of cells in DDi and how do they relate to the EOD rate (sensory sampling), the fish's location and its back-scanning movements across landmarks.

## Results

*Figure 2* shows the wireless transmitter, the tetrode and their assembly prior to (A–C), and after implantation (D). Extracellular recordings from neurons within the dorsal pallium (dorsal pallium (DD) were wirelessly transmitted as the fish freely swam in a large circular tank that contained differently shaped landmarks (*Figure 2E*, see Materials and methods). We aimed our tetrodes to the intermediate subdivision of DD: DDi (see Methods, *Figure 2—figure supplement 1*). *Figure 3A* shows an example extracellular recording together with the raster plot for the four units that could be sorted based on their shape. *Figure 3B* shows the shape of the sorted units and their first three principle components that were used for clustering units (see Materials and methods). The Electric Organ Discharge rate (EODr, bottom red trace, *Figure 3A*) was calculated using the EOD signal that was simultaneously recorded by the tank electrodes (see Materilas and methods).

Below, in the first section, we will describe the overall spiking activity of DDi cells and its relation to swimming and EODr. Increases in EODr were previously demonstrated to occur both near objects (food and landmarks, *Jun et al., 2016*) but also independently of objects (*Jun et al., 2014b*). We thus go on, in the second section, to first quantify the temporal relation of DDi cell spikes and variables such as swim speed, EODr and EODr per unit distance traversed (sampling density) independently of objects. We then proceed (third section) to focus in on the important connection between DDi cell spiking and objects and the tank boundary. Gymnotiform fish make active sensing movements near landmarks and food (*Jun et al., 2016*; *Nelson and Maciver, 1999*). In the fourth section, we therefore focus on the critical connection between DDi cell spiking and active sensing movements.

### Spikes were sparse and occurred mainly during movement

*Figure 4A* shows the distribution of the average firing rates of all units recorded from all the fish and across all trials, highlighting sparsity of spiking in this region (25 units, five fish, 23 trials, 20 hr and 36 min of recording). We found that that units in DDi fired at strikingly low rates, and that it was very unlikely to encounter a spike at low swimming speeds or during periods of quiescence. *Figure 4B* shows the histogram of swim speeds calculated across all time bins and trials (purple) and those in time bins where there was a spike (green). Notice a peak at low swim speeds in the purple

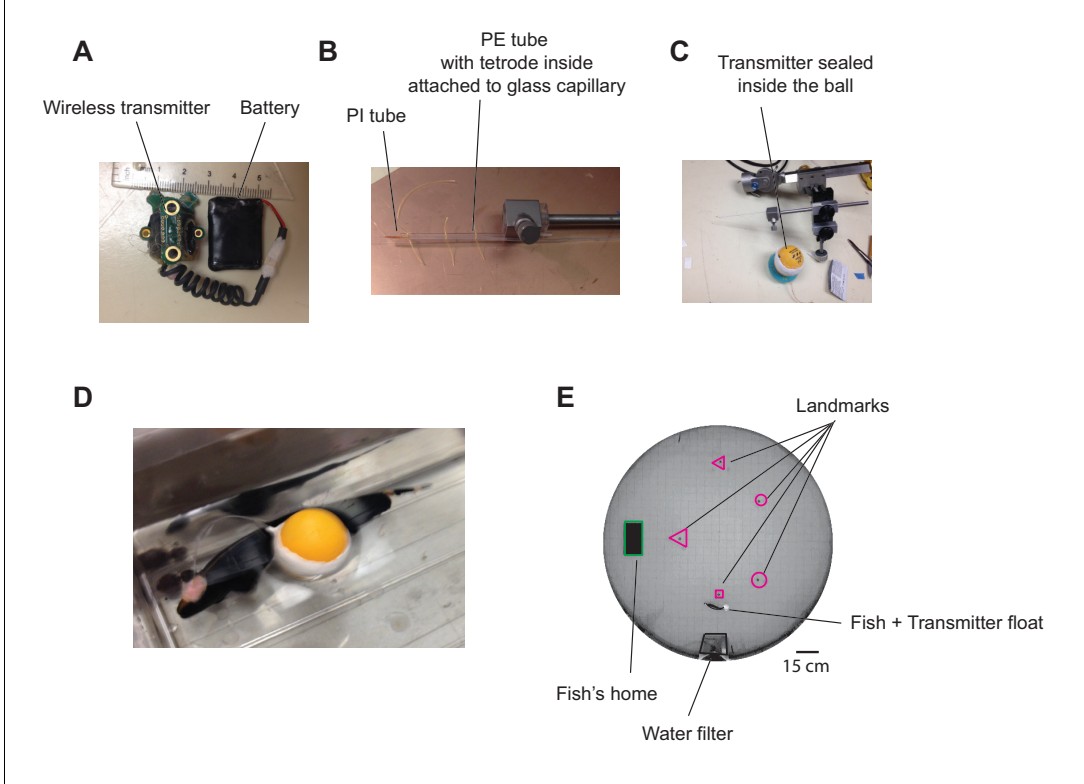

**Figure 2.** Experimental setup and example wireless recording. (A) Wireless transmitter system used for recordings. (B) A long tetrode was constructed and mounted on an electrode holder to be attached to a micromanipulator. (C) The other end of the tetrode was connected to the transmitter, and the ensemble was placed in a ping-pong ball and sealed. (D) Pictures of a fish after tetrode implantation together with the transmitter float (picture from a related species with the same size). (E) Recordings were performed in a large experimental tank containing various landmarks made with clear and opaque plexiglass, as well as a water filtering system.
DOI: https://doi.org/10.7554/eLife.44119.003

The following figure supplement is available for figure 2:

**Figure supplement 1.** Cresyl violet stained section through the pallium of an implanted fish illustrating the location of the tetrode.
DOI: https://doi.org/10.7554/eLife.44119.004

histogram, which is missing in the green one. It is known that at very low speeds, that is during moments of behavioral quiescence or 'down-states', the EOD rate is lower than when the fish is active (generally less than 50 Hz down states in *Gymnotus sp.*, *Jun et al., 2014b*), resulting in a reduction in sensory sampling rate. Consistently, we found that spikes were highly unlikely during down-states (*Figure 4C*, notice the absence of spikes at EODr <50 Hz) indicating that spikes in DDi neurons are likely linked to movement and the ongoing electrosensory sampling of the environment.

## Relation between spiking activity, EOD rate, swim speed and sampling density

The EODr continuously varies in time (*Figure 3*, red trace). Higher EODrs indicate higher sensory acquisition rates, and large, fast transients in the EODr has been previously reported to occur near landmarks in the context of spatial learning (*Jun et al., 2016*). Fish's swim speed is variable and is also associated with proximity to landmarks and changes during learning (*Jun et al., 2016*). Sampling density, the number of EOD pulses per unit length, depends on both EODr and swim speed and therefore also varies strongly with the fish's location near landmarks (*Jun et al., 2016*). Below, we describe the relation between DDi spiking and EODr, swim speed and, consequently, with sampling density.

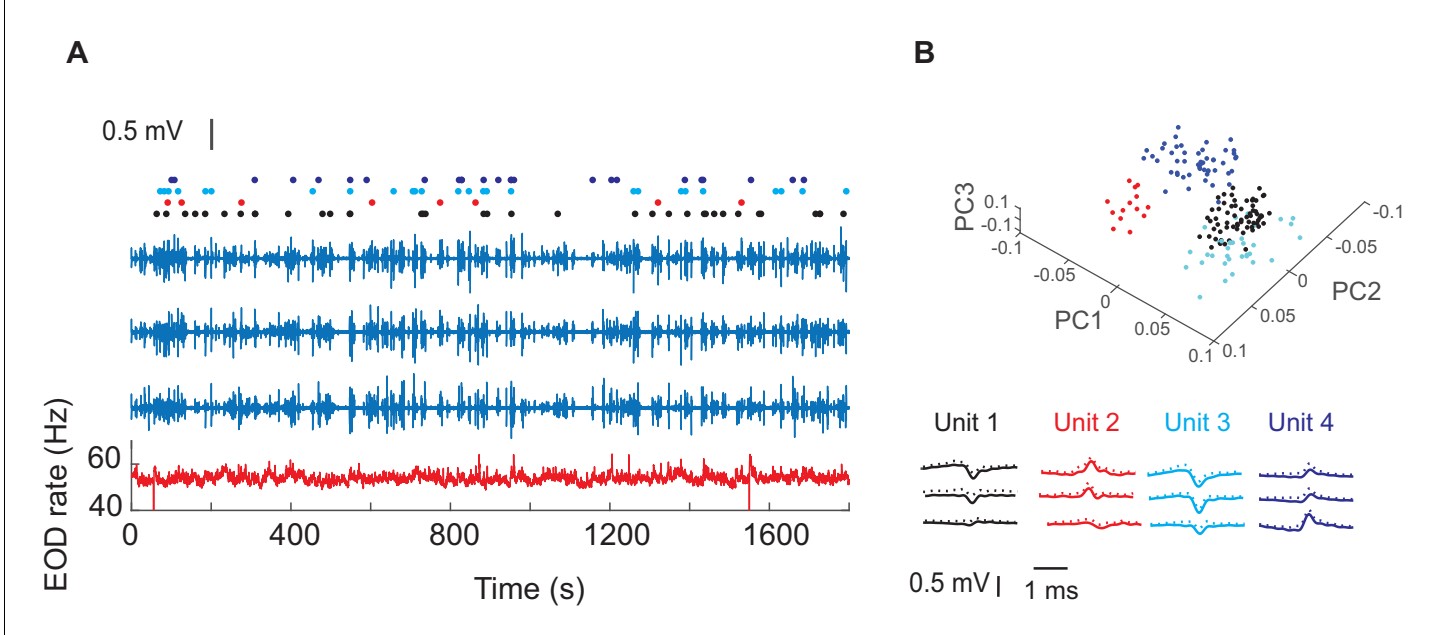

**Figure 3.** Example recordings in one fish and four isolated units. (A) The blue traces show extracellular recordings on the three of the four tetrode channels after EOD spike removal (see Materials and ethods). The fourth channel was used as reference. The red trace shows instantaneous EOD rate calculated based on EOD recordings obtained by electrodes inside the experimental tank. Raster plots show spike timing corresponding to the four units. (B) Average (SD) of waveforms of the isolated units from this recording and their first three principle components (PCs).

DOI: https://doi.org/10.7554/eLife.44119.005

## EODr

To examine the relation between sensory acquisition rate and spiking activity of DDi units, we calculated spike-triggered-EODr (stEODr) averages for all units that fired more than 10 spikes in a trial (see an example in *Figure 5A*). We found that for the majority of units (15 out of 21), the mean EODr in an 8 s window around the spike time was significantly higher than that calculated for randomly time-shifted spikes (*Figure 5A* black trace, see Materials and methods). Moreover, we found that for many units the mean EODr increased prior to spike time and peaked within a few hundred

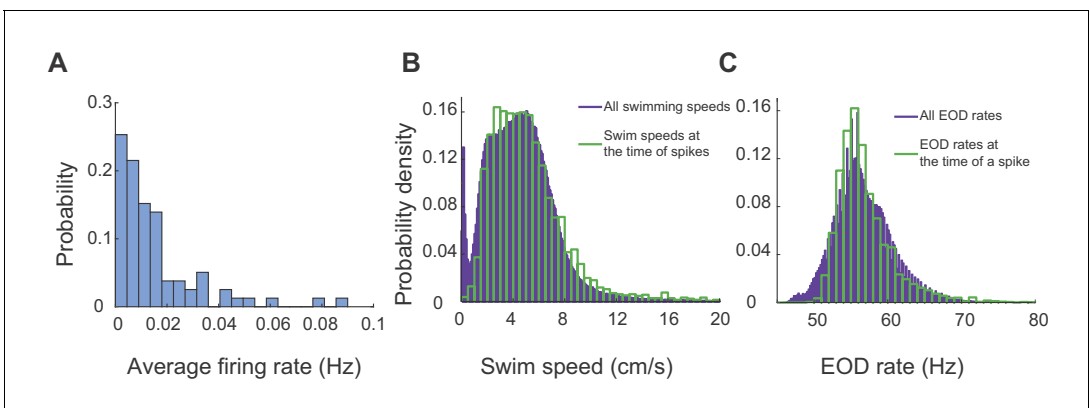

**Figure 4.** DDi units spiked sparsely during periods swimming and active sensing. (A) Probability histogram of average firing rates observed in all units and all trials (25 units, 23 trials, five fish). (B) Probability density function of all observed swim speeds (purple bars), and swim speeds at the time of a spikes (green bars). Spikes were unlikely at very low swimming speeds. (C) Probability density function of all observed EOD rates (purple bars) and those observed at the time of spikes. (green bars). Spikes were unlikely at low EOD rates (<50 Hz) corresponding to down states.

DOI: https://doi.org/10.7554/eLife.44119.006

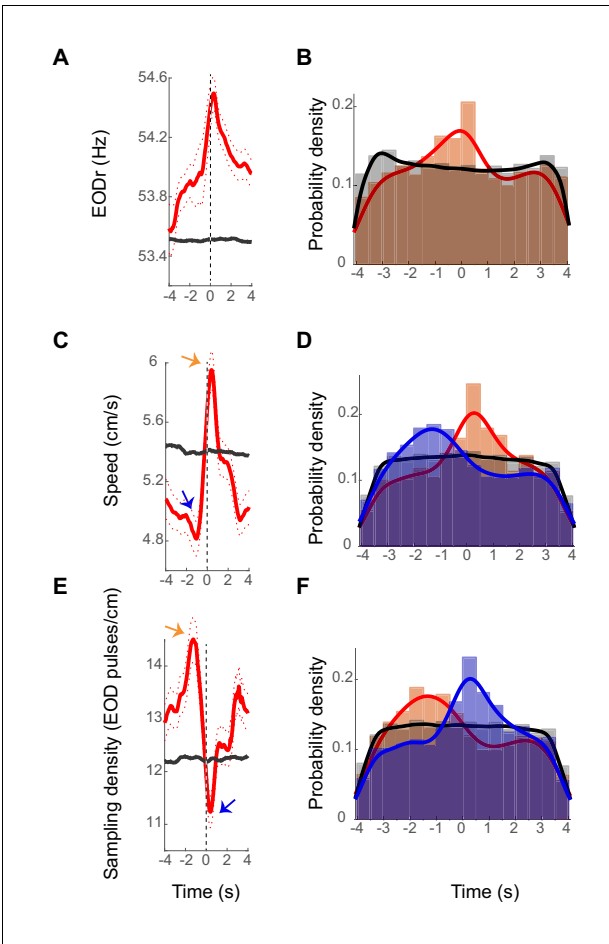

**Figure 5.** Examples stEODr, stSpeed and stSmpD averages. (**A**) StEODr average for an exemplar unit (solid red curve, dotted curve: standard error). The black curve shows the stEODr average for 100 random time shifts of the same spike train. The dashed vertical line corresponds to the spike time (zero). (**B**) Probability density of timing of stEODr peaks around individual spikes for units with significant peak in stEODr average (orange bars, 13 units in five fish) and for 100 random time shifts of the same spike trains (gray bars). There was a clear peak of stEODr around zero that did not exist in the random data. (**C**) StSpeed average (red curve) for the same unit shown in A, and the corresponding average for randomly shifted spike times (black curve). Arrows point to the dip and peak in the stSpeed pre- and post- spike respectively. (**D**) Probability density of speed dips (blue) preceding the spike and peaks (orange) after spikes for units with a significant peak in stSpeed average (14 units in five fish, gray: randomly time-shifted spike trains). (**E**) stSmpD average for the same unit. stSmpD showed a clear peak before the spike and sharply decreased immediately post-spike, with the minimum occurring after spike time. (**F**) Probability distribution of timing of the peak (orange) and dip (blue) in sampling density for individual spikes of units with significant peaks in both stEODr and stSpeed shows the same pattern of a pre-spike increase followed by a post-spike decrease (10 units in five fish). Fits in all histograms are non-parametric fits with Gaussian kernels.

DOI: https://doi.org/10.7554/eLife.44119.007

The following figure supplements are available for figure 5:

**Figure supplement 1.** stEOD, stSpeed and stSmpD averages for all units and all fish 1 (red curves).

DOI: https://doi.org/10.7554/eLife.44119.008

**Figure supplement 2.** stEOD, stSpeed and stSmpD averages for all units and all fish 2 (red curves).

DOI: https://doi.org/10.7554/eLife.44119.009

**Figure supplement 3.** stEOD, stSpeed and stSmpD averages for all units and all fish 3 (red curves).

DOI: https://doi.org/10.7554/eLife.44119.010

**Figure supplement 4.** stEOD, stSpeed and stSmpD averages for all units and all fish 4 (red curves).

DOI: https://doi.org/10.7554/eLife.44119.011

**Figure supplement 5.** stEOD, stSpeed and stSmpD averages for all units and all fish 5 (red curves).

DOI: https://doi.org/10.7554/eLife.44119.012

*Figure 5 continued*

**Figure supplement 6.** Comparison of average spike-triggered EODr for spikes of a unit when they occurred in close vicinity of landmarks or tank boundary (<3 cm, red curves) and when they occurred far from them (>10 cm away, black curves).
DOI: https://doi.org/10.7554/eLife.44119.013
**Figure supplement 7.** Relation between EOD rate and swimming speed.
DOI: https://doi.org/10.7554/eLife.44119.014
**Figure supplement 8.** Comparison of the shape of spike-triggered averages across trials with the overall average for fish 1.
DOI: https://doi.org/10.7554/eLife.44119.015
**Figure supplement 9.** Comparison of the shape of spike-triggered averages across trials with the overall average for fish 2.
DOI: https://doi.org/10.7554/eLife.44119.016
**Figure supplement 10.** Comparison of the shape of spike-triggered averages across trials with the overall average for fish 3.
DOI: https://doi.org/10.7554/eLife.44119.017
**Figure supplement 11.** Comparison of the shape of spike-triggered averages across trials with the overall average for fish 4.
DOI: https://doi.org/10.7554/eLife.44119.018

milliseconds around the time of the spike. 13 out of the 21 units showed a significant peak in the stEODr average.

*Figure 5A* shows an example unit with a significant peak in its stEODr average (stEODr for all the other units are presented in *Figure 5—figure supplements 1–5*). *Figure 5B* shows the probability density histogram of timing of peaks in EODr relative to individual spikes from all units and all fish (orange bars). The peaks were more likely to occur around the timing of the spike; within a second prior to and 0.5 s after spike time (median peak time was zero). The histogram for the timing of stEODr calculated using randomly time-shifted spike trains did not show such a peak around the spike time (gray bars, see Materials and methods). Interestingly, we found that the amplitude of the average stEODr was often higher when the fish was in the close vicinity of landmarks or tank boundary (<3 cm away) compared to when it was far from them (>10 cm away, *Figure 5—figure supplement 6*). However, for most units, the peak in the average stEODr could also occur when the fish was far from all landmarks. The coupling between DDi spikes and EODr transients could therefore occur at locations away from landmarks, albeit less strongly. Presumably, such associations of DDi spikes and apparently spontaneous increases in active sensing reflect as yet unknown functional linkages between the brainstem EOD control network (*Comas and Borde, 2010*) and pallium.

## Swim speed

We next considered the correlation between swim speed and spiking of the units. Interestingly, we observed that the fish's spike-triggered-swim-speed (stSpeed) often decreased to a minimum (dip) before the time of spike, followed by a peak after (*Figure 5C*, blue and orange arrows, respectively. see also *Figure 5—figure supplements 1–5*). Fourteen of 21 units showed a significant peak in stSpeed average after the spike time. Ten of these 14 units also had a significant peak in their stEODr average, with all stEODr average peaks except one occurring after the spike time (*Figure 5—figure supplement 5* , 1st unit). *Figure 5D* shows the distribution of the timing of speed dips (blue bars) and peaks (orange bars) across all units and fish (median stSpeed dip time = −0.56 s median stSpeed peak time = 0.2 s). Unlike the stEODr, the timing of the dips and peaks in stSpeed were significantly shifted to negative and positive values that is before and after the timing of the spike, respectively (for dips: $p=4.4\ e^{-24}$, for peaks: $p=8.5e^{-11}$, non-parametric sign test). This pattern was not evident for the timing of peaks and dips of stSpeed calculated for randomly time-shifted spikes (gray bars show the peak times, similar pattern was observed for the dips, data not shown). We tested whether the link between spikes and EODr and speed could be due to inherent correlations that exist between EODr and swim speed irrespective of the presence of spikes. EODr is indeed lower during down-states when the swim speed is at its minimum and we found that they were correlated during swimming as well (*Figure 5—figure supplement 7*). The relationship between the

two, however, could not be fully explained with a linear function. In three out of 5 fish, less than 13% of the variance in EODr could be explained by the variance in speed. We therefore conclude that although EODr and Speed could be correlated in some instances, they are also under independent control, and the correlation between spikes and these two variables could not be solely due to the inherent correlation between the two.

## Sampling density

We next used units that showed significant peaks in both stEODr and stSpeed averages (10 units) and calculated the spike-triggered sampling density as the ratio of stEODr to stSpeed (stSmpD, exemplar unit is shown in *Figure 5E*, see *Figure 5—figure supplements 1–5* for all units). The sampling density has units of EOD pulses emitted per unit length (pulses/cm), it is a measure of electrosensory spatial acuity, and has been previously proposed to be an indicator of spatial attention (*Jun et al., 2016*). Mirroring stSpeed, we found that for most units the stSmpD average showed a peak before spike time and a dip after it (*Figure 5F* orange and blue arrows, respectively). *Figure 5F* shows the probability distribution for the peaks (orange bars) and dips in stSmpD (blue bars) for all units and all fish (median stSmpD peak time = −0.56 s, median stSmpD dip time = 0.24 s). The timing of the peak and dip in stSmpD were not significantly different from the timing of dip and peak in stSpeed, and occurred significantly before and after spike time, respectively (for peaks: p=2.68 e$^{-24}$, for dips: 1.1 e$^{-10}$, non-parametric sign test). This pattern was not present in the timing of dips and peaks calculated for stSmpD of randomly time-shifted spikes (gray bars show the peak times, similar pattern for the dips, data not shown).

In the case of all three variables, the timing of peaks and dips relative to individual spikes were variable (relatively wide distribution of peak/dip times evident from the histograms shown in *Figure 5B*). This result suggests that there is no causal 1–1 relationship between peaks/dips of these variables and triggering of DDi or vice-versa. This is perhaps not surprising as there is no reason to believe that fluctuations in for example EODr that occur at a much higher rate than spiking rate of DDi neurons should causally relate to individual spikes we recorded from the small group of DDi cells. Despite the large degree of variability in the dynamics of EODr and speed around individual spikes, however, we found a clear bias in the timing of EODr peak and swimming speed and sampling density peaks and dips relative to the timing of DDi spikes: spikes were most likely to occur after a dip in swim speed, with EODr often increasing prior to a spike and peaking around the time of the spike. The combination of the pre-spike decreasing swim speed and increasing EODr resulted in a strong increase in SmpD before the spike. Similarly, the post-spike increasing swim speed and decreasing EODr resulted in a strong SmpD dip post-spike. The changes in swim speed were clearly the dominant factor contributing to this change in SmpD. This is completely consistent with the results of *Jun et al. (2016)*, which shows that sampling density increases in the vicinity of landmarks. The net effect of these coordinated changes in EODr and swim speed leads to strongly enhanced electrosensory sampling (SmpD) over the trajectory traversed ~1–3 s before a spike followed by reduced sampling for up to 3 s after the occurrence of the spike.

## Spiking occurred near boundaries and landmarks

*Figure 6* shows examples of swimming trajectories, spiking locations, and firing rate maps for eight units recorded in five fish. Units in DDi tended to fire in multiple locations within the experimental tank often close to the tank boundaries and landmarks (*Figure 6—figure supplement 1*). To quantify place-specificity of each unit's activity we calculated the 'place information' in bits/spike (see Materials and methods) for the 21 units that fired more than 10 spikes during the trial. The average of information across all units and trials was 1.58 bits/spike (SD = 1.15). For each unit and trial, we tested the statistical significance of place information level by comparing it to that calculated from randomly time-shifting the spike train relative to swim trajectory. Examples of the firing pattern of units that conveyed significant place information, and those that did not are given in *Figure 6A and B*, respectively. Eight of the 21 recorded units in four fish (3,1,2,2 units in each fish respectively) showed significant place specificity. Interestingly, five out of eight units with significant place selectivity also had significant peaks in their stEODr averages (*Figure 5—figure supplements 1–5* and *Figure 6—figure supplement 2*). Three of these units (two fish) further showed a clear peak in SmpD before spike time. There was no clear correlation between lack of place specificity and

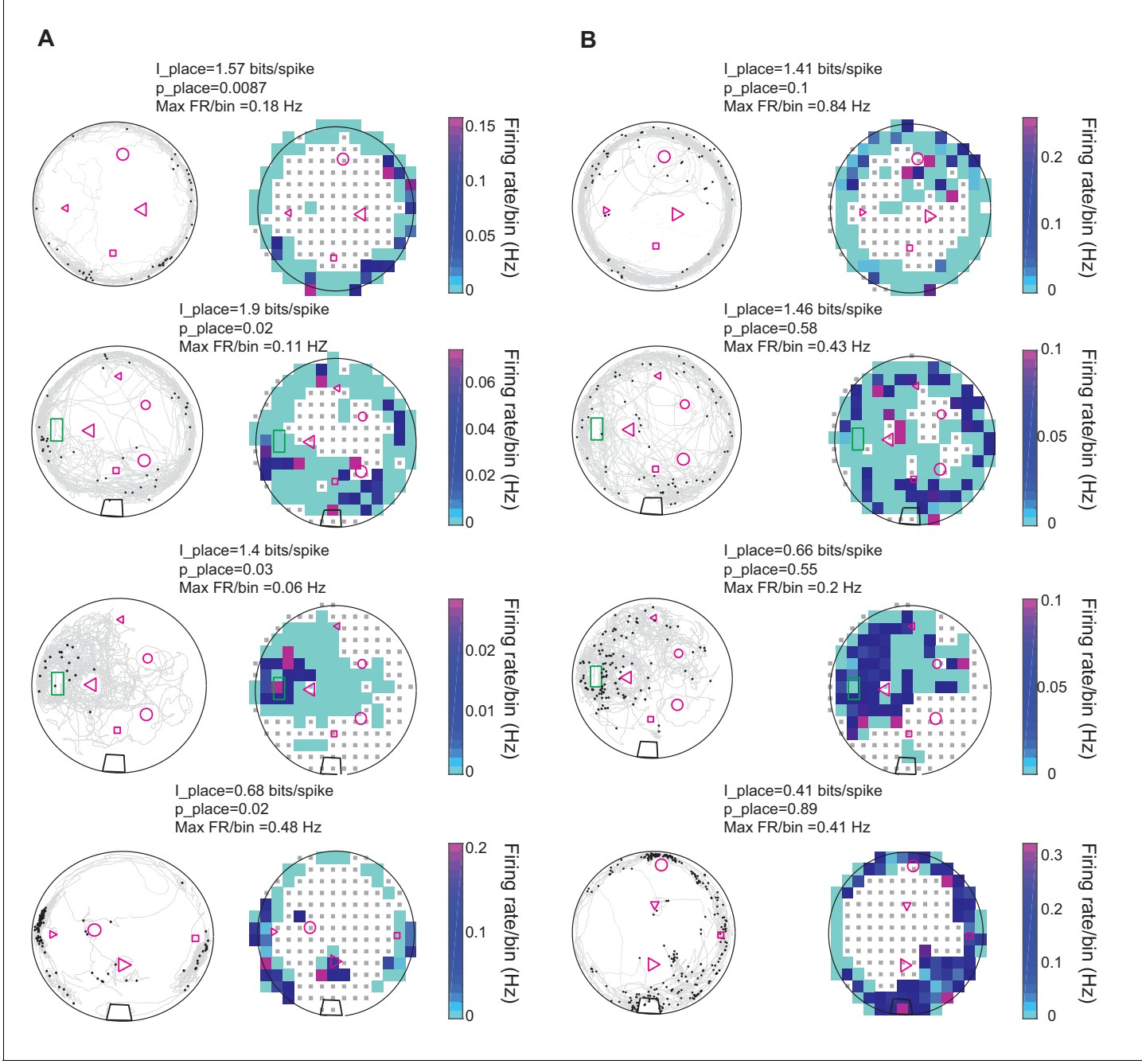

**Figure 6.** Spatial properties of DDi units.  (A) Left columns: Examples of spiking patterns of four units in four fish that conveyed significant place information. gray curves show fish's swimming trajectory, and black dots represent spikes. Right column: Firing rate maps of the same units shown to the left. Firing rate per bin is calculated by dividing the total number of spikes by the time spent in that bin. Only bins where the fish visited more than five times are used for the plot. The range of the color plot was clipped to the 97th percentile of the data for visual representation purposes (see Materials and methods). The value of the maximum firing rate/bin (Max FR/bin) is indicated above the plot. The place information for the unit and its level of significance compared to randomly time shifted spike trains are indicated as I_place and p_place, respectively. (B) Same as A but for units that did not show statistically significant place specificity. The green rectangle corresponds to fish's home, the black trapezoid in the bottom three plots show the location of a water filter. Other shapes represent various landmarks placed in the tank. Note that the fish could go inside the home area, but not other landmarks. Small gray squares denote bins excluded from analysis due to visit counts less than 5 (see Materials and methods).
DOI: https://doi.org/10.7554/eLife.44119.019

The following figure supplements are available for figure 6:

**Figure supplement 1.** Comparison of the total number of spikes fired by each unit within a trial around landmarks or tank boundary and away from them.

*Figure 6 continued on next page*

*Figure 6 continued*

DOI: https://doi.org/10.7554/eLife.44119.020

**Figure supplement 2.** Significance levels for place information and the peak amplitude for the stEODr average for each of the 21 units recorded in five fish (1- p values).

DOI: https://doi.org/10.7554/eLife.44119.021

**Figure supplement 3.** Comparison of the overall firing rate within and across trials for fish 1.

DOI: https://doi.org/10.7554/eLife.44119.022

**Figure supplement 4.** Comparison of the overall firing rate within and across trials for fish 2.

DOI: https://doi.org/10.7554/eLife.44119.023

**Figure supplement 5.** Comparison of the overall firing rate within and across trials for fish 3.

DOI: https://doi.org/10.7554/eLife.44119.024

**Figure supplement 6.** Comparison of the overall firing rate within and across trials for fish 4.

DOI: https://doi.org/10.7554/eLife.44119.025

**Figure supplement 7.** Comparison of the overall firing rate within and across trials for fish 5.

DOI: https://doi.org/10.7554/eLife.44119.026

absence of a peak in stEODr (8 out of the 13 units that were not place specific, showed a significant peak in their stEODr average, and other five did not, *Figure 6—figure supplement 2*).

Interestingly, changes in landmark configurations often resulted in changes in the firing patterns of the units and gain or loss of significance in conveyed place information. *Figure 7A* shows examples of the effect of changes in landmark configuration on spiking activity of three units. *Figure 7B* shows a summary plot of the effect of landmark removal on the firing rate of the units in a 10 cm radius around the location of the landmark (16 trials and 10 units, 12 out of 16 trials showed a reduction). For example, removing landmarks often resulted in a reduction in firing rate of the unit around the location of the missing landmark. Remarkably, in one case where the landmark was moved to a new location, the unit shifted its preferred firing rate to the new location of that same landmark (*Figure 7A–iii*, small triangle). Further, comparison of the firing rates in the two conditions normalized to the largest rate showed that the rate was significantly reduced with landmark removal (*Figure 7B*, inset). Landmark removal was not the only manipulation that resulted in change of firing rate around a given landmark. Adding landmarks to the tank, which initially only contained the home structure resulted in the reduction of firing rate of a few neurons around the home structure despite it being still present (*Figure 7—figure supplement 1*). Therefore, we conclude that a subset of DDi units fire in a location- specific manner and that their spatial specificity is tightly linked to the presence of landmarks.

## Swim direction and relative landmark location preference

*Gymnotus* can swim in both forward and backward directions. Backward swims are often observed in the context of prey capture and spatial navigation and serve to bring prey or landmarks that the fish has swam past, back near the fish's head and rostral trunk (*Pedraja et al., 2018*; *Jun et al., 2016*; *Nelson and Maciver, 1999*), the regions containing the highest density of electroreceptors (*Carr et al., 1982* and *Castelló et al., 2000*).

Interestingly, we found that most units in DDi showed preference for spiking during backward swims. *Figure 8A* shows the distribution of the swim direction preference indices (see Materials and methods) for all units in all fish. The distribution showed a significant bias toward negative values that correspond to backward swims (mean (SD) = −0.196 (0.33), p=0.0072, nonparametric sign test).

We next asked, do units in DDi have a preferred landmark location, that is are they more likely to fire when a landmark has a specific location relative to the body of the animal? To answer this question, we calculated a probability map for landmark locations when a given unit was active. To do this, we divided spike-triggered landmark locations by position-triggered landmark locations (see Matrials and methods, *Figure 8B*). *Figure 8C* shows normalized landmark probability at the time of spiking of four exemplar units. For each unit, we then calculated an anterior-posterior and a left/right preference index (see Materials and methods). *Figure 8D* shows preference vectors plotted using these preference indices for all units in all fish. Vectors corresponding to exemplar units in *Figure 8C* are indicated within this plot. We found that for most units the probability of landmark

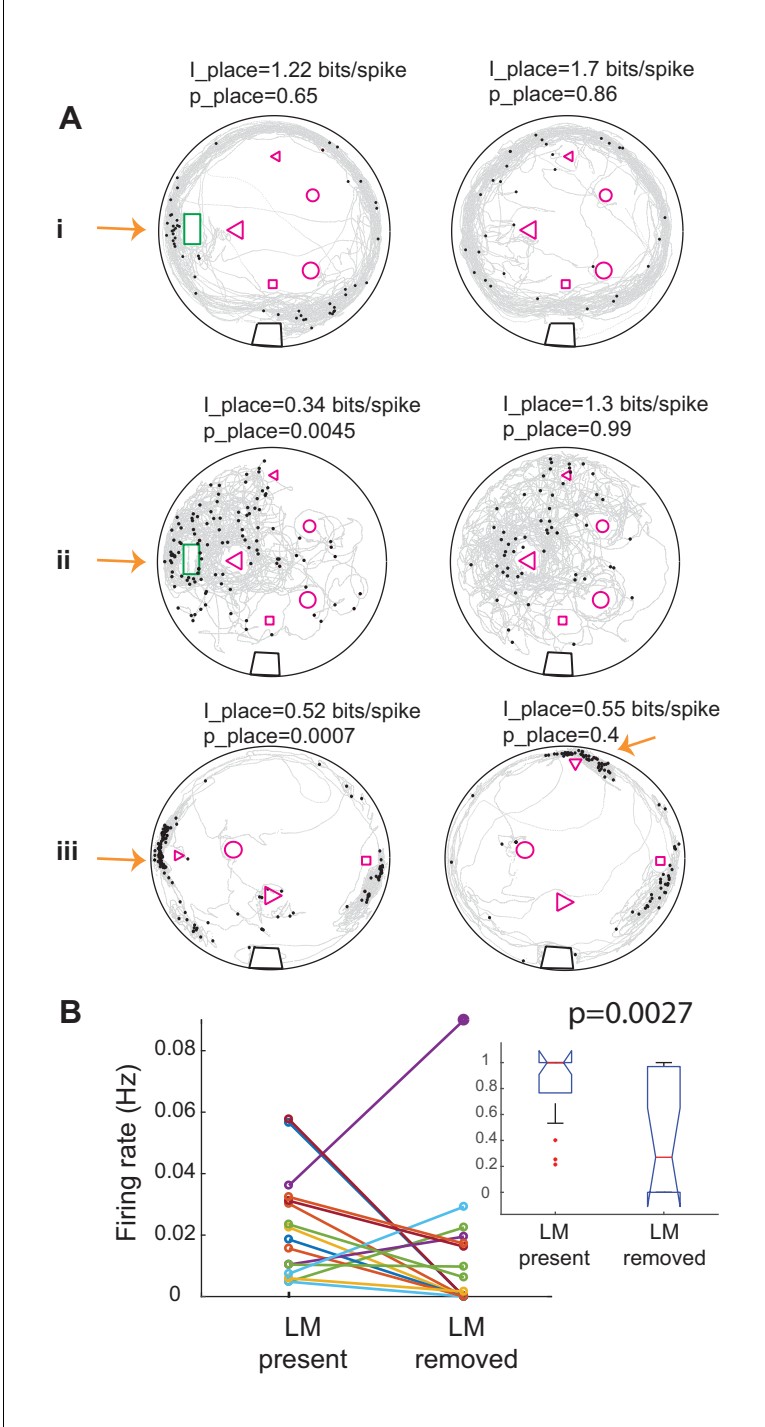

**Figure 7.** Removing landmarks often resulted in reduction of firing rate near the missing landmark. **(A)** Examples of the effect of removing or displacing a landmark in three units in three fish. In i and ii the home was removed and in iii the small triangle was moved to a new location. Note that spikes now occur near the triangle in its new location. Place information and its significance level compared to randomly time-shifted spikes are shown above each panel. Arrows point to the location of the moved landmark. Note that the unit shown on the left panel ii was significantly less place specific than chance. **(B)** Summary plot of the effect of landmark removal on the firing rate of 10 units in three fish, 16 trials. In 12 out of 16 trials, the firing rate decreased near the removed landmark. Inset: comparison of the normalized firing rates between the two conditions for the same data set. Firing rate was significantly lower in after landmarks were removed. Kruskal-Wallis p value is indicated on the plot.

DOI: https://doi.org/10.7554/eLife.44119.027

*Figure 7 continued on next page*

*Figure 7 continued*

The following figure supplement is available for figure 7:

**Figure supplement 1.** Effect of change in landmark configuration on the firing rate of units from two fish around the location of home (within 10 cm).

DOI: https://doi.org/10.7554/eLife.44119.028

presence was higher in the region posterior to the body of the fish (*Figures 8E*, 14 out of 21 units). This bias was not present for left- versus right location for the landmarks (*Figures 8F* and 10 out of 21 units preferred spiking to contralateral objects). Because in all fish the tetrode was implanted in the left lobe of the pallium, these results indicate that activity in DDi can be driven by bilateral input; this is likely due to the strong commissural connections between the left and right DD (*Giassi et al., 2012c*). In summary, most DDi units we recorded spiked in response to objects located in the region posterior to the body of the fish as fish was swimming backwards. These units therefore appear to correlate to a scanning movement that would lead to relocating these objects from the trunk to the head and more anterior parts of the body, which contain the highest density of electroreceptors.

## Discussion

In this study, we wirelessly recorded and characterized neural activity within the dorsal pallium (DDi) of freely swimming pulse-type weakly electric fish *Gymnotus sp.* This is the first study to report on location and movement related response properties of cells in a region of a teleost fish's pallium that has similar connectivity, and may be homologous to, the CA3 area of mammalian hippocampus (*Elliott et al., 2017*, see *Figure 1* and Introduction). A previous study (*Elliott and Maler, 2015*), using a related (immobilized) gymnotiform fish (*Apteronotus leptorhynchus*), found that, in the absence of sensory input, DDi cells were completely silent. In response to electrocommuncation and acoustic signals, DDi cells discharged very sparsely, at long latencies and with a small number of spikes. These results are comparable to our data. The physiological data of *Elliott and Maler (2015)* are also consistent with an earlier report on the induction of immediate early gene expression in DDi neurons after *Apteronotus*' were presented with electrocommunication signals (*Harvey-Girard et al., 2012*). Further work will be required to determine whether different DDi cells respond to electrolocation versus electrocommunication signals.

Below, we use the results of *Wallach et al. (2018)* to interpret our results. Wallach et al recorded the responses to object motion from the thalamus (PG) of a curarized *Apteronotus* leptorhynchus. Our anatomical studies (*Giassi et al., 2012b*; *Giassi et al., 2012c*; *Elliott et al., 2017*) indicate that object motion responses in DDi neurons must be driven by PG (via DL, see *Figure 1* and Introduction). Two questions must be answered to validate the conclusions we present below. First, can we use responses recorded from one species of gymnotiform fish, *Apteronotus* leptorhynchus, to interpret the results of derived from a different gymnotiform fish, *Gymnotus sp.*? Secondly, can the responses of PG cells to an object moving near a stationary fish (*Apteronotus*) be used to interpret the response of DDi cells when the fish are moving near a stationary object (*Gymnotus*)?

The electrosensory circuitry of the hindbrain (electrosensory lobe) is nearly identical in *Apteronotus* and *Gymnotus* (*Lannoo and Maler, 1990*; *Shumway, 1989*). The midbrain electrosensory structures (TS, tectum), their connections to PG and pallium and pallial circuitry are also nearly identical in both species (*Giassi et al., 2012a*; *Giassi et al., 2012b*; *Giassi et al., 2012c*). Gymnotiform fish utilize two active sensing behaviors. Pulse type fish (e.g *Gymnotus sp.*) will increase their EODr and sampling density near landmarks and food (*Jun et al., 2016*); *Apteronotus* is a 'wave type' fish and maintains a constant EODr when foraging. Importantly, both Gymnotus (this paper; *Jun et al., 2016*; *Pedraja et al., 2018*) and *Apteronotus* (*Nelson and Maciver, 1999*) will use the same back scanning movements (B-scans) in proximity to salient landmarks. We conclude that the high degree of circuitry and behavioral (active sensing) similarity across these species permit data from *Wallach et al. (2018)* to be used in the interpretation of our data.

The responses of DDi cells might, in principle, be derived from both sensory input and a corollary discharge from a motor brain region that initiates and directs movement (*Crapse and Sommer, 2008*; *Straka et al., 2018*). In this case, the responses of DDi cells to object motion (*Wallach et al., 2018*) might be different from those in response to fish motion past objects (this paper and

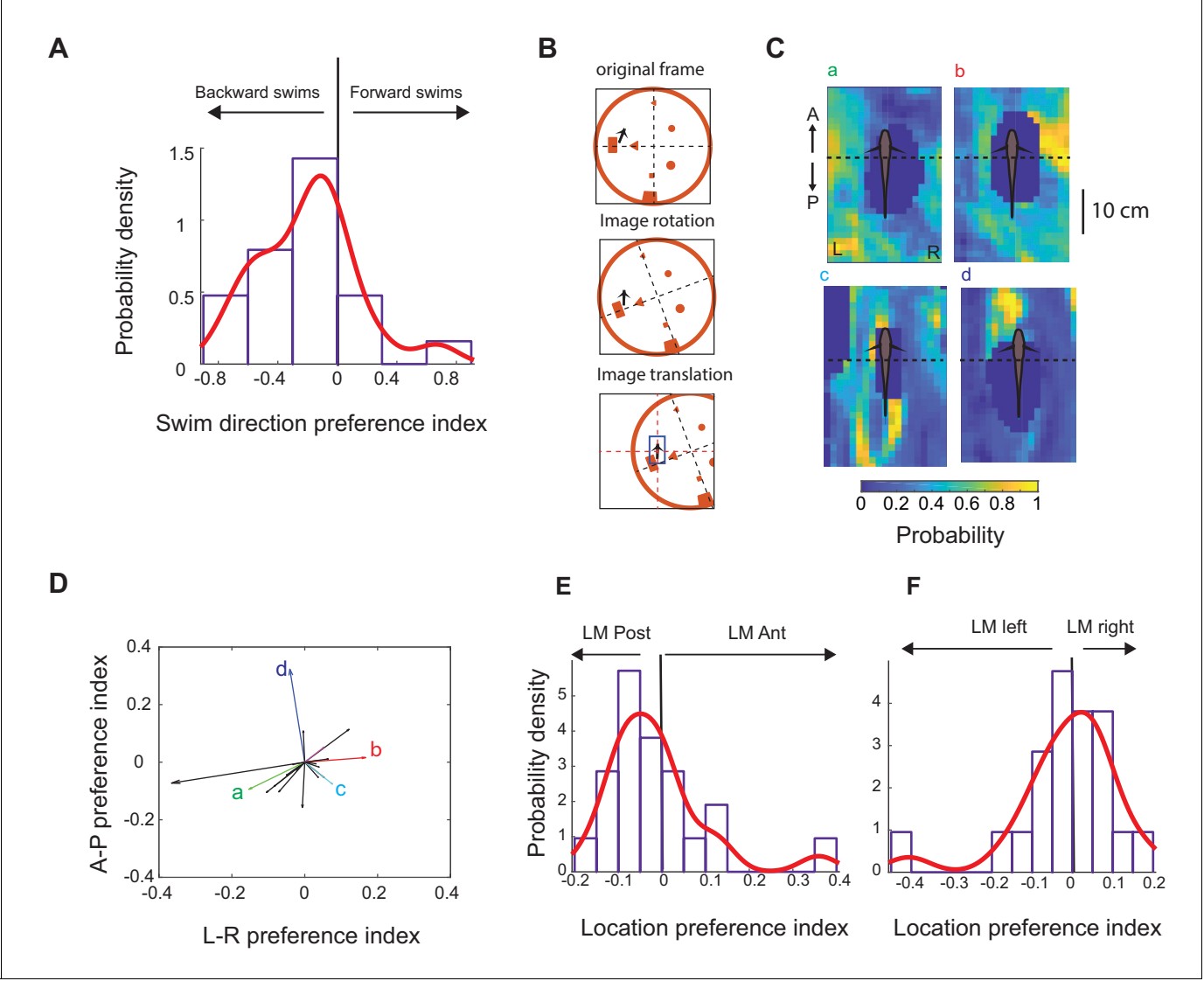

**Figure 8.** Relationship between swimming direction and preferred landmark location. (A) Probability distribution of direction preference index (21 units, five fish). Negative values correspond to preference for spiking during backward swims, with the maxima ± 1 corresponding to spiking only during forward or backward swims, respectively. The red curve shows a non-parametric fit with Gaussian kernels. There was a significant bias for spiking for negative swim direction preference indices (p=0.0072, non-parametric sign test). (B) Schematics of the procedure for calculating Spike and Location Triggered Landmark matrices (STLM and PTLM, respectively). To calculate the spike-triggered landmark (STLM) matrix, first a 160 × 120 element matrix corresponding to the absolute location of landmarks and tank boundary as viewed from the video recording was constructed. Matrix elements that contained landmarks or tank boundaries were set to 1 (depicted as orange on the diagram), whereas other elements were set to zero. Next, at the time of each spike of a given unit, this matrix was rotated and then translated such that fish's position at that time was centered in the matrix and its head was facing north. The matrix sum was then calculated for all spikes from a given unit. Similarly, a PTLM matrix was calculated as the sum of such rotated and translated landmark matrices calculated for all time points during the trial (down-sampled fish location, see Materials and methods). STLM was then divided by PTLM to calculate the probability of presence of landmarks around the fish (within 10 cm, blue rectangle). The pixel values within this 10 cm wide window were normalized to the maximum pixel value to generate the probability plots. (C) Examples of landmark presence probability plot at the time of spikes of various units. Units b and c are from the same fish and other units are from three other fish. Dark blue regions around the fish correspond to locations that were either never occupied by any landmark or occurred very rarely (lower 10 percentile of all PTLM values) and were excluded from the analysis (see Materials and methods). The anterior part of the fish was defined as the anterior 1/3 of the animal, which corresponded to the zero value on the A/P axis. The L-R axis was defined as the left or right side of the fish, with the fish's mid-line serving as zero. The left - right preference index was calculated as the difference between the maximum probability on the left side and the right side divided by the sum of the two. We similarly calculated an anterior-posterior preference index. (D) Vector plot of the direction preference indices for anterior-posterior (AP) and left-right (LR) preference indices. Small letters next to the arrows correspond to exemplar units shown in panel C. (E) Most units showed preference for landmarks that were located on the posterior 2/3 of the body (blue shaded area). (F) Left- right preference indices were equally likely to be positive or negative.

*Figure 8 continued on next page*

*Figure 8 continued*

DOI: https://doi.org/10.7554/eLife.44119.029

*Jun et al., 2016*). Our anatomical studies (*Giassi et al., 2012b*; *Giassi et al., 2012c*; *Elliott et al., 2017*) show that the central dorsal pallium (DC) provides the only pallial output that can initiate or drive movement (via the tectum, see *Figure 1*). Most importantly, DC does not project to either DDi or DL. We conclude that, in gymnotiform fish, there is no direct pallial corollary discharge pathway associated with movement, and that the DDi responses to electrosensory motion signals will therefore likely be similar for object versus self-motion.

In the discussion below, we first summarize our most important results and then interpret them with respect to neuroanatomy (effectively identical in *Apteronotus* and Gymnotus, *Giassi et al., 2012a*; *Giassi et al., 2012b*; *Giassi et al., 2012c*) and the results presented in *Jun et al. (2016)*. Critical to our interpretation is that our experiments were carried out in fish that were learning about a new environment -the particular spatial layout of the tank (open maze) environment. We therefore compare our data to the 'early learning' results of *Jun et al. (2016)*, where the fish also first encountered landmarks in the same open maze environment and, by using active sensing, learned to identify landmarks. We also incorporate the recent results of *Wallach et al. (2018)*, who examined the response of *Apteronotus* PG neurons to looming/receding and longitudinal object motion. We note that our forward object motion stimuli are, given the lack of corollary discharge, equivalent to the backward swimming component of B-scans (*Figure 1*). Similarly, backward object motion is equivalent to the forward swim direction.

We found that DDi units only fired when the fish was moving and the EODr was high (*Figure 4*, Up state: *Jun et al., 2014b*). Discharge typically occurred at multiple locations distributed within the experimental tank and often near tank boundaries and landmarks where the fish spent most of its time (*Figure 6* and *Figure 6—figure supplement 1*).

Spikes from a third of the units we recorded conveyed significant spatial information (*Figure 6*). The spatial pattern of spiking and its spatial specificity changed when the landmark configuration was altered (*Figure 7A*, *Figure 7—figure supplement 1*). We found that in the majority of units, spikes were linked to the presence of landmarks and that removing landmarks resulted in a reduction in the firing rate in the location of the missing landmark (*Figure 7B*, *Figure 7—figure supplement 1*). These findings demonstrate that cells in DDi encode the presence of landmarks or boundaries, albeit in a stochastic manner, as spikes were not fired at every instance the animal visited a given location.

Neural mechanisms underlying spatial navigation has been intensely studied for decades in mammals and, especially, in rodents (*Eichenbaum, 2017*; *Moser et al., 2017*; *Chersi and Burgess, 2015*). Various cell types in different areas of the hippocampal formation have been identified to encode various self, environmental, and social cues that enable the animal to successfully navigate toward a goal. These cells include place cells (*O'Keefe and Nadel, 1978*; *Moser et al., 2017*), boundary/border cells (*Lever et al., 2009*; *Savelli et al., 2008*; *Solstad et al., 2008*), grid cells (*Moser et al., 2017*; *Savelli et al., 2017*), head direction cells (*Taube, 2007*; *Peyrache et al., 2017*), goal direction cells (*Sarel et al., 2017*), and more recently social place cells (*Danjo et al., 2018*; *Omer et al., 2018*). There are some similarities between the responses of DDi cells and hippocampal place and boundary cells in that place and boundary cells also fire sparsely (*Diamantaki et al., 2016*; *Hainmueller and Bartos, 2018*; *Rolls, 2016*) and only when the animal is moving (*Chen et al., 2013*; *Winter et al., 2015*; *Song et al., 2005*; *Harvey et al., 2018*), and their discharge is often associated with the presence of a landmark (*Muller and Kubie, 1987*). However, the responses of DDi cells are clearly not exact matches of either place or boundary/border cells. Boundary (subiculum, *Lever et al., 2009*) and border (entorhinal cortex, *Solstad et al., 2008*) cells respond to specific locations at a maze edge, while the DDi cells typically responded at several sites. A recent study (*Vinepinsky et al., 2018*) reported that, in the goldfish DL, a subset of neurons (named 'border cells') responded strongly when the fish were near the border of the experimental enclosure; it is not clear from their description whether these cells were active during movement or rest (or both). These cells are clearly different from both mammalian border cells and the DDi cells we recorded in that they discharged at all border locations. It may be that the classic spatially tuned neurons of the

mammalian hippocampus (plus entorhinal and subicular cortices) have not yet been sampled in the pallium (DL, DD or DC) of teleost fish. Alternatively, the teleost pallium may employ different mechanisms to encode the location of the fish with respect to environmental features.

Our most important result is that the firing rates of many DDi cells are linked to the types of active sensing (EODr and sampling density increases and B-scans) that occur when the fish is learning the spatial layout of the open maze environment (*Jun et al., 2016*). We found that over the whole recording session, for many units on average, the spike-triggered EODr average started increasing prior to spike time and continued till slightly after the spike (13 out of 21 units, *Figure 5A,B*). In many cells (14 out of 21 units), the spike-triggered speed showed a dip prior to a spike and dramatically increased post-spike (*Figure 5C,D*). This combination resulted in an increased sampling density before and during the spike followed by its reduction post-spike. *Wallach et al. (2018)* report that many PG cells fire as an object approaches the fish (*Apteronotus*). We hypothesize that, in *Gymnotus*, the increased EODr and sampling density will also strongly drive PG spiking and therefore DL spiking activity. The only sensory input to DDi is from DL which leads us to conclude that DDi spikes will likely feedback (via DDmg) to DL while DL is activated by PG input. We further discuss this point below.

Gymnotiform fish exhibit stereotyped forward/backward swimming motions (B-scans) that are important for learning about landmarks (*Pedraja et al., 2018*; *Jun et al., 2016*). Further, a recent study has shown that the fish must perform exploratory movements (e.g. B-scans) near objects in order to perceive their shape (*Fujita and Kashimori, 2019*). *Jun et al., 2016* demonstrated that EODr and sampling density are highest during the backward phase of B-scans suggesting that PG will be very strongly driven by this phase of B-scans. Remarkably, *Wallach et al. (2018)* described PG units that responded strongly and specifically throughout the forward movement of an object, that is the same relative motion that would occur during backward swimming. We found that most units we recorded (17 out of 21) showed a preference for spiking during the backward swimming (*Figure 8A*). Moreover, we found that most units (12 out of 17) with preference for spiking during backward swims spiked most when objects were initially located near the trunk region (*Figure 8C, D*). In other words, these units spiked during a movement that would result in the object ending up near the head – the 'foveal' region (*Figure 1*). We hypothesize that this PG activity drives DDi cells (via DL) during back-swimming. The DDi spikes will feedback (via DDmg) to DL while it still being activated by its strong ongoing PG input. Under these conditions, there will be at least three temporally overlapping sources of excitatory synaptic input to DL cells: (1) PG spiking driven by the backward scanning of the landmark, (2) DL spiking driven by its local recurrent connectivity (*Trinh et al., 2016*), and (3) feedback input from DDi (via DDmg). DL is likely the site for storage of spatial memory. We therefore hypothesize that the spatial learning described by Jun et al is driven by synaptic plasticity at one or more of these synaptic inputs to DL neurons.

Active sensing movements may be a common strategy employed during spatial learning in at least teleost fish and mammals. Mormyrid fish have independently evolved an active electric sense and at least one species (Gnathonemus pertersii) uses a variety of stereotyped movements during electrosensory recognition of objects (*Hofmann et al., 2014*; *Schumacher et al., 2016*). In fact, B-scans were first identified in mormyrid fish (named 'va-et-vient' movements, *Toerring and Moller, 1984*). Somewhat similar results have been reported for a blind cave fish that uses lateral line input generated by active swimming past landmarks for learning about the spatial organization of their environment (*Burt de Perera, 2004a*; *Burt de Perera, 2004b*). It would be important to determine whether teleosts with good vision (e.g. goldfish) actively direct their gaze (via body or saccadic eye movement) when learning the spatial layout of their environment. Remarkably, a study of place cells in blind rats concluded that 'blind rats tended to make exploratory contacts with the objects more often than did sighted rats' and might use such contacts for spatial information (*Save et al., 1998*).

Rodents use head-scans as 'a spatially directed investigative behavior' (*Poulter et al., 2018*); there is a clear functional analogy between head-scans and B-scans. The most direct evidence for the role of active sensing movement and spatial learning comes from an important recent study. *Monaco et al. (2014)* showed that headscans drive formation or strengthening of place field discharge of hippocampal cells of rats (*Monaco et al., 2014*). Under the conditions of this study, it is likely that visual input was the main input driving place field formation during headscans. Additional evidence in favor of this idea comes from a study that showed that active self-movement was required for normal grid cell firing patterns (*Winter et al., 2015*).

We hypothesize that, during B-scans, DDi feedback to DL will also drive the formation of place/object- associated discharge of DL cells, and that this is a key element of spatial learning in gymnotiform and, possibly, mormyrid fish. More generally, we propose that active sensing movements are important for generating a representation of spatial relations in the dorsal telencephalon of vertebrates. Future experiments that record from, for example pallium or hippocampus, of freely moving animals, combined with accurate tracking of active sensing movements (e.g. head, whisker and body movements, saccades etc.) may prove valuable for determining precisely what is 'learned' during active spatial learning.

## Materials and methods

### Wireless transmitter and tetrode fabrication

A wireless transmitter/receiver system (*Figure 2A*, TBSI-W16, Triangular Biosystems Intl, Durham, NC) was used for transmitting and receiving neural recordings in freely swimming fish. Tetrodes were constructed using 12-micron stablohm wires (stablohm 650, California Fine Wire Company, Grover Beach, CA) and wound using a Neuralynx Tetrode Spinner (Neuralynx, Bozeman, MT). Each tetrode was made 15 cm long and passed through slightly shorter flexible polyethylene tubing (PE, PE10, 0.61 mm OD x 0.28 ID, Warner Instruments Corp, Hamden, CT) such that either end of the tetrode was sticking outside the tube (*Figure 2B*). The one end of the tetrode to be implanted was further passed through a 0.5 cm long 180-micron diameter polyimide tubing (PI, Microlumen, 068-I, Lot#24331) and fixed in place using a mixture of Krazy glue and dental cement (Jet denture repair powder, Lang Dental Mfg. Co Inc, Wheeling, IL) for additional rigidity (*Figure 2B*). The PE tubing containing the tetrode was attached to a glass capillary, which could be solidly inserted inside an electrode holder attached to a micro-manipulator (*Figure 2C*). Melted Polyethylene glycol (PEG) was used to attach the tubing to the glass capillary. At room temperature, PEG solidifies and acts as a water-soluble glue. Once the electrode was implanted, the tubing could be separated from the glass capillary by gently pouring water on the solidified PEG. The other end of the tetrode protruding from the PE tubing was also stabilized inside the PE tubing using another piece of PI tubing and glue. Individual tetrode wires were then separated at that end and attached to the input ports of the electrode interface board (EIB) which could then be plugged in to the transmitter. One of the four electrodes was used as reference. At this point, the electrodes were first electroplated using Neuralynx gold plating solution and then with a solution of Ethylene Dioxythiopene monomer (EDOT) and Polystyrene sulfonate (PSS) using nanoZ plating protocols and software (MultiChannel Sytems MCS GmbH, Germany). Individual tetrode impedances varied between 100 and 200 kOhms. The EIB was then attached to the transmitter-battery ensemble and they were then put inside a cut-open Ping-Pong ball, which was used as a float for the transmitter system. A few pieces of vibration absorbing gel (Z8006, Kyosho, Lake Forest, CA) were added inside the ball and around the transmitter system to prevent movement and vibrations that could result in transmission noise. The Ping-Pong ball was then closed back (*Figure 2C*) and water proofed using mouldable glue (Sugru, London, UK).

### Animal preparation and electrode implantation

All animal procedures were performed in accordance with the regulations of the animal care committee of the University of Ottawa. *Gymnotus sp.* of either sex were used for all experiments. Before implanting, tetrodes were first cleaned by submersion in 70% ethanol for 15 min and then in sterile saline solution (0.9% NaCl) for another 15 min to rinse off the ethanol. They were then further sterilized using UV illumination. Fish were anesthetized in a small container with tricane methanesulfonate (MS-222; Aqua Life, Syndel Laboratories) in tank water solution. They were then transferred to a holder outside of water where their head and body could be stabilized in preparation for surgery and electrode implantation. The MS-222 solution in oxygenated deionized (DI) water was continuously administered in this setup through a tube that was inserted in the mouth, and the fish's body was covered with wet sponges and Kimwipes to protect the skin. Before opening the skull, it was first completely dried, then some crazy glue was applied on the contralateral side of the planned implant to make the skull surface rough. This procedure helped with the final closure step as the dental cement mixture could better adhere to the remaining pieces of roughed bone. Additionally, a few indentations were made using a dentist drill on the surface of the contralateral skull. These

indentations served as extra attachment points as they were filled with the dental cement mixture at the closing stage. The dorsal pallium was exposed by a small craniotomy and the tetrode was micro-manipulated to the DD region.

DD of gymnotiform fish is divided into superficial (DDs), intermediate (DDi) and magnocellular (DDmg) sub-regions (*Figure 1*, *Giassi et al., 2012a*). *Elliott et al. (2017)* suggested that DDi, on the basis of its connectivity pattern, was similar and perhaps homologous to the CA3 hippocampal field. We directed our tetrodes towards DDi using the data of *Giassi et al. (2012a)*, an Atlas of the *Gymnotus* brain (*Corrêa et al., 1998*) and a high-resolution lab atlas of the *Gymnotus* brain for guidance. This atlas has previously successfully directed tracer injections precisely into DDi (*Giassi et al., 2012c*; *Elliott et al., 2017*). Surface sulci delimiting DD from the more medial dorsomedial pallium and more lateral dorsolateral pallium were used to guide the electrodes to DD and at a site rostral to the very caudal DDmg. The penetration depth was adjusted to pass through DDs but remain confined to DDi (*Figure 2—figure supplement 1*).

Once the electrode was implanted, the opening of the skull was covered with small pieces of thin plastic sheets (0.5 mil thick FEP film made with Teflon fluoroplastic, CS Hyde Company, Lake Villa, IL). The sheets were then secured to the rest of the skull using a UV curable sealant (Aegis Pit and Fissure Sealant, Keystone Industries, Germany), which was also used to seal any other small remaining openings. After curing with UV light, a mixture of dental cement and Krazy glue was used to further seal the opening and to cover the incision areas on the fish's skin. At this point, the MS-222 solution was slowly diluted with DI water and the PE tubing containing the tetrode was released from the glass capillary by applying a gentle flow of water. Once the fish resumed breathing, it was gently taken out of the setup and transferred to the experimental tank and allowed to recover overnight (*Figure 2D*).

Recordings were obtained the next day in four out of five fish (Fish 1–4) and on day 4 after surgery in one fish that took longer to recover (Fish 5). The recordings were obtained as long as the battery lasted. This was between 5 and 6 hr in three out of five fish. For the fish that took a long time to recover (Fish 5, i.e. to resume normal activity level and to swim consistently around the tank) we could only record for about an hour during the fourth day before the battery ended, as we had attempted recording the days before, when the fish was not swimming reliably. For fish 3, we have a little over 2 hr of recording after which the fish pulled out the implant and had to be sacrificed right after. At the end of a recording session, we deeply anaesthetized the fish and perfused it in a standard manner (*Giassi et al., 2012a*). We attempted to remove the tetrode and extract the brain with minimal damage and to then section and stain (Cresyl Violet, *Giassi et al., 2012a*) in order to identify the tetrode location. We were successful in locating the tetrode track in two fish (*Figure 2—figure supplement 1*).

## Experimental setup and trials

The experimental tank was 1.5 m in diameter and fish were tested in shallow water (~10 cm) to facilitate video tracking by restricting fish's swimming trajectory in two dimensions (*Figure 2E*). A full description of the tank construction and the landmark shapes can be found in (*Jun et al., 2014a*; *Jun et al., 2016*). The length of the tetrode (15 cm) was chosen such that the wireless transmitter float would not exert any force on fish's head. Landmarks were made of acrylic and were secured to the bottom of the tank using suction cups and could be added or removed. Recordings were done as the fish were learning about a new environment. Each trial lasted between 30 min – 1 hr during which the landmark configuration was stable. Landmarks were sometimes removed or displaced to test the effect on the firing properties of the units. This was done with in presence of the fish. Electric Organ Discharge (EOD) signals were captured using four pairs of graphite electrodes (Mars Carbon 2 mm type HB, Staedler, Germany) attached to the tank walls at equal spacing (*Jun et al., 2014a*). All experiments were performed in the dark and the animal's behavior was monitored under IR illumination using a camera (C910, Logitech, IR filter removed) that was mounted above the tank. The camera acquired images at 1600 × 1200 resolution and had a frame rate of 15 Hz.

## Wireless data reception and spike sorting

Analog signals received at the receiver were digitized using (CED mkII, Cambridge Electronic Design, UK) and further analyzed using CED's Spike2 software. EOD signals sensed by four pairs of tank electrodes were also acquired simultaneously using the CED acquisition system. Neural recordings also contained spikes from the EOD. To facilitate spike sorting, EOD spikes were removed from the recording offline, by setting the neural recording trace in the time-window −2.8:2.8 ms around each EOD spike to zero. We estimated the number of missed spikes as a result of the EOD pulse removal procedure during periods of high EODr in two fish. In each fish, we took a window of 250 s long during which EODr was above average and calculated the total number of spikes (detected by setting a threshold on the extracellular recordings) once using the raw recording and next using the EOD-spike-removed trace. We found that indeed some spikes were lost in the process of removing EOD spikes. The number of spikes detected from the raw trace and EOD-spike-removed trace were 28 and 25, respectively in one fish, and 192 and 168, respectively in the other. Therefore, in periods of heightened EOD rate, we could potentially loose around 10% of the spikes. It is important to note that this number corresponds to the total number of spikes and puts an upper limit on the percentage of lost spikes for individual units. Therefore, the sparsity in firing of DDi neurons is not simply a result of the EOD-removal procedure.

*Figure 3A* shows an example recording: the three blue traces are extracellular recordings from the three tetrode channels after removing EOD spikes. and the red trace shows instantaneous EOD rate calculated based on tank electrodes. Spike sorting was done using Spike2 software based on spike waveform shape. The threshold for spike detection was set high and kept constant for all trials in one fish, therefore, only units with high signal-to-noise ratio were kept for subsequent analysis. Initial sorting by shape was followed by fine tuning spike clusters using principle component analysis and visual inspection of individual spikes (*Figure 3B*). Due to low firing rate of units, clusters were sometimes not completely separable and therefore some units may be considered as multi-unit.

## Analysis of EOD rate – spike relationship

The following data analysis were all performed in MATLAB (MathWorks Inc). Spike-triggered EOD rate (stEODr) and speed (stSpeed) averages were calculated in an 8 s window around the timing of each spike (±4 s). Spike-triggered sampling density (stSmpD) was calculated by dividing stEODr by stSpeed for each spike. For each unit, then the spike-triggered averages were calculated using all spikes fired by that unit across all trials. The results were qualitatively the same when performed separately for each trial (*Figure 5—figure supplements 8–11*). The same procedure was repeated 100 times for spikes circularly shifted by a random time (by at least 30 s and at most the length of the trial minus 30 s) for comparison. The average of stEODr over the entire 8 s window for all spikes was compared to that calculated for the stEODr for randomly time shifted spikes using Kruskal-Wallis test. To calculate the significance of the peak in stEODr and stSpeed averages, the value of the stEODr and stSpeed for each spike at the time of the peak in the average was calculated. These values were then compared to average EODr and speed calculated over the whole window for all spikes using the Wilcoxon sign rank test. Histograms shown in *Figure 5B* quantify the variability in the timing of peaks/dips in stEODr, stSpeed and stSmpD relative to individual spikes. To calculate these probability density functions, the timing of the largest peak in EODr, Speed or SmpD was measured for each spike. To calculate the histogram of timings of the dip in stSpeed and stSmpD, the timing of the smallest local minimum was used. The same procedure was repeated for individual spikes from the randomly time shifted data set. This distribution is shown in gray in *Figure 5B*. The timing of the peaks was randomly distributed for randomly time shifted dataset, whereas it showed a clear bias for the actual data.

## Analysis of fish's swimming trajectory and spatial firing rates

Position, and swimming direction of the fish in the experimental tank was calculated using custom software written in MATLAB as well as those available from Ty Hedrick's lab (*Hedrick, 2008*; *Fotowat et al., 2019*). To calculated spatial firing properties of the units, the area of the experimental tank was divided into 16 × 16 cm bins and the total number of spikes fired in each bin was divided by the time-spent in that bin. Bins that had less than five visits during the whole trial were not included in the analysis. A visit was counted when the fish's head first arrived at a given bin. If

the fish stayed within a bin for more than one frame (video frame rate = 15 Hz), visit number was still counted as 1, and was allowed to increase only when the fish left the bin and returned to it another time. For visualization purposes, color range shown in the firing rate map plots was clipped at 97 percentile of firing rate of that unit across all bins. This was done to avoid bins where the fish spent a very small amount of time to saturate the color plot. The maximum firing rate per bin is indicated above these plots (*Figure 6*).

To assess whether the firing rate of units were stationary within and across trials, we divided each trial in two halves (between 15–30 min long), each half consisted of 5 bins within which the value for firing rate was calculated. We then compared the five values obtained for the first and second half of the trial using non-parametric Kruskal-Wallis test (*Figure 6—figure supplements 3–7*). Additionally, we compared the values of the firing rate calculated over all 10 bins for each trial with those from the consecutive trial using the same statistics.

Spatial information in bits per spike was calculated using the firing rate maps as described previously (*Skaggs et al., 1996*; *Rubin et al., 2014*):

Spatial information (bits/spike)= $\sum p_i \left(\frac{r_i}{r_m}\right) log2 \left(\frac{r_i}{r_m}\right)$

Where $p_i$ is the probability of the animal being in the $i$th bin, calculated as the ratio of time spent in bin i divided by the total trial time, $r_i$ is the firing rate in bin $i$ and $r_m$ is the mean firing rate for that unit. Only bins with more than five visits were included in the analysis. To calculate statistical significance of spatial information, this information measure was recalculated for 1000 randomly time shifted spike trains (conserving the relative timing among spikes) superimposed on the same swimming trajectory. The spatial information conveyed by a unit was considered significant if it exceeded 95th percentile of the distribution of the information calculated for the randomly shifted spike trains (t-test was used to assess significance). This procedure was done separately for each trial during which the landmark configuration was stable, and the firing rate of the units was largely stationary (*Figure 6—figure supplements 3–7*). Units that conveyed significant information for one or more trials were counted as spatially specific units (For units/trials that showed significant within-trial change in firing rate random time shifting of spikes would not be appropriate for assessing significance. None of these units showed significant place information using this measure). Due to sparse firing of neurons, if the firing rate of a unit was higher at a certain time period (this could be for example due to visiting a certain location only at that time), randomly time-shifting the whole spike train would preserve this bias and yield high information levels even for the time-shifted data. Using this method for testing the significance of place information, therefore, provides a conservative estimate for the p values. Indeed, when we recalculated the significance of place information, this time using randomly shuffled spike times (as opposed to randomly time shifting the whole train) we found that more units showed significance in their place information (11 as opposed to 8 units). In this paper, we have reported the results using the more conservative method that conserves relative spike timing to report the number of place-specific units.

To examine the effect of removing a landmark on the firing rate of a given unit we calculated the average firing rate for the bins that were within a 10 cm radius of the landmark, before and after its removal.

## Swimming direction and landmark location preference analysis

Spike rate during forward (backward) swim was calculated as the number of spikes that occurred during forward (backward) swim divided by the total time spent swimming forward (backward). Because of the small number of spikes, we pooled forward (backward) turns with forward (backward) swims. Direction preference index was calculated as the difference of the firing rate during forward swims and backward swims divided by their sum. Negative values of the direction preference index indicated preference for spiking during backward swims. To calculate the spike-triggered landmark (STLM) matrix, first a 160 × 120 element matrix corresponding to the absolute location of landmarks and tank boundary as viewed from the video recording was constructed (each matrix element corresponded to a 10 × 10 pixels area in the video recording). Matrix elements that contained landmarks or tank boundaries were set to one whereas other elements were set to zero (*Figure 8B*, original matrix). Next, at the time of each spike of a given unit, this matrix was rotated and translated such that fish's position at that time was centred in the matrix and its head was facing north (*Figure 8B*, matrix rotation/translation). The matrix sum was then calculated for all spikes from a given unit.

Similarly, a Position- triggered Landmark (PTLM) matrix was calculated as the sum of such rotated and shifted landmark matrix calculated every 1.33 s (the fish's position data, which had a resolution of 15 Hz, was down-sampled 20 times). STLM was then divided by PTLM and then normalized to its maximum to calculate the 'normalized probability of presence of landmarks' at a given location and this was further characterized within a window extending up to 10 cm around the fish (*Figure 8B*, blue rectangle). This distance was chosen as a conservative upper range for object detectability with the electric sense based on *Jun et al. (2016)*.

Elements of PTLM matrix that were zero or very small (smaller than lower 10 percentile of all matrix elements) indicate either the lack or very small number of instances when a landmark was present at that position relative to the body during a given trial. For each unit, these positions were excluded from the analysis as they would result in very large values (or infinity in case of zero PTLM) in the STLM/PTLM matrix that are not due to large number of spikes (i.e. large STLM value) but due to zero or limited number of encounters (very small PTLM). In the exemplar probability plots shown in *Figure 8C* these are the dark blue areas.

We divided the area around the fish into four regions: left and right side of the fish, each of which was further divided into two regions: anterior 1/3 and posterior 2/3 of the fish's body. The region located around the anterior third of the fish, corresponds to the regions near fish's head and upper trunk (gill region) that contains the largest number of electroreceptors (*Carr et al., 1982*; *Castelló et al., 2000*). We then calculated a left - right preference index as the difference between the maximum probability on the left side and the right side divided by the sum of the two. We similarly calculated an anterior-posterior preference. Positive left- right (anterior-posterior) preferences corresponded to locations to the right side (anterior) of the fish. For each unit, we then calculated a preference vector with its x and y value equal to the left- right and anterior posterior preference indices, respectively (*Figure 8F*). For each unit, we used the amplitude and direction of this vector as an indicator of the strength and orientation of preferred landmark location for that unit. Because in all fish the tetrode was implanted on the left hemisphere, landmarks to the left side of the fish were ipsilateral to the location of the electrode and those to the right of the fish were contralateral.

## Acknowledgements

We thank Florian Engert for his helpful comments, suggestions and support for this manuscript. We also thank Bill Ellis for technical support and Erik Harvey-Girard, Armin Bahl, Martin Haesemeyer and Roy Harpaz for their helpful suggestions. This research was supported by NSERC grants 04336 and CIHR grant 153143 to LM.

## Additional information

### Funding

| Funder | Grant reference number | Author |
|---|---|---|
| Natural Sciences and Engineering Research Council of Canada | 04336 | Len Maler |
| Canadian Institutes of Health Research | 153143 | Len Maler |

The funders had no role in study design, data collection and interpretation, or the decision to submit the work for publication.

### Author contributions

Haleh Fotowat, Conceptualization, Resources, Data curation, Software, Formal analysis, Supervision, Validation, Investigation, Visualization, Methodology, Writing—original draft, Project administration, Writing—review and editing; Candice Lee, Formal analysis, Investigation, Writing—review and editing; James Jaeyoon Jun, Conceptualization, Resources, Software, Methodology; Len Maler, Conceptualization, Resources, Supervision, Funding acquisition, Validation, Writing—original draft, Project administration, Writing—review and editing

## Author ORCIDs
Haleh Fotowat http://orcid.org/0000-0003-0372-4912
Len Maler http://orcid.org/0000-0001-7666-2754

## Ethics
Animal experimentation: All animal procedures were performed in accordance with the regulations of the animal care committee of the University of Ottawa, protocol number CMM-2897.

## Decision letter and Author response
Decision letter https://doi.org/10.7554/eLife.44119.034
Author response https://doi.org/10.7554/eLife.44119.035

## Additional files

### Supplementary files
• Transparent reporting form
DOI: https://doi.org/10.7554/eLife.44119.030

### Data availability
Data sets and analysis files have been deposited in University of Ottawa's Institutional repository.

The following dataset was generated:

| Author(s) | Year | Dataset title | Dataset URL | Database and Identifier |
|---|---|---|---|---|
| Fotowat H | 2019 | Neural activity in a hippocampus-like region of the teleost pallium is associated with active sensing and navigation | https://dx.doi.org/10.20381/38902 | University of Ottawa's Institutional Repository, 10.20381/38902 |

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
