## [Decision Letter]

Thank you for submitting your article "Neural activity in a hippocampus-like region of the teleost pallium are associated with navigation and active sensing" for consideration by *eLife*. Your article has been reviewed by three peer reviewers, one of whom is a member of our Board of Reviewing Editors, and the evaluation has been overseen by Timothy Behrens as the Senior Editor. The following individual involved in the review of your submission has agreed to reveal his identity: Maurice J Chacron (Reviewer #2).

The reviewers have discussed the reviews with one another and the Reviewing Editor has drafted this decision to help you prepare a revised submission.

Summary:

This is a highly interesting study, where the authors present the first recordings from the forebrain of a freely swimming weakly electric fish. Here the authors present recordings from neurons within the forebrain and analyze their responses with respect to EOD rate, spatial position, swimming speed, and active sensing movements. Their results suggest that the activity of pallium neurons are modulated by EOD rate as well as swim speed and spatial location. The use of wireless recording of neural signals in freely moving fish is exciting. Such approaches are absolutely necessary for us to understand how these animals acquire spatial memories and use their electric sense to navigate within their environments. Nevertheless, reviewers felt that the manuscript would greatly benefit from more quantitative analyses of the data and better articulation of the findings in the context of pre-existing literature. These suggestions are noted below.

Essential revisions:

The initial part of the manuscript shows that DDi spikes are sparse, occur only during swimming and active sensing, and are preceded by increased EOD sampling density (Figures 3 and 4). The later part of the manuscript presents data to argue that DDi cells detect landmarks and swim directional and positional preferences. But these two aspects are not sufficiently interwoven to generate an understanding of the role of DDi neurons during active sensing. For neurons that have significant stEODr and landmark specificity, what exactly would these neurons be coding for? More quantitative analyses looking at causality as well as interactions between EOD rate and other variables (position, swim speed, etc.) are needed to better assess what is causing spiking in these neurons. Further, they should test whether different variables (e.g., swim speed and EOD rate) are correlated, as this would then trivially explain why some neurons respond to changes in both variables.

The statistics used assume stationary data. However, this is unlikely to be the case for the dataset considered, as these are naive fish that are discovering a new environment. The authors need to assess nonstationarities in their data and test how this will affect their results. If there are nonstationarities, the random-shifted spike trains are likely to have different statistics than the original one.

The reviewers felt that the evidence that this is a place-like response related to spatial navigation based on path integration is quite weak. However, evidence presented in the manuscript convincingly shows that DDi spiking is related to an active sensing based detection mechanism, and encodes landmark locations based on EOD and B-scans. The authors allude to path integration in the Introduction and Discussion, and draw a comparison between DDi firing and hippocampal place cell responses. Spatially specific firing related to landmarks is not sufficient evidence for path-integration based navigation, and indeed, spatially specific firing is seen in many mammalian brain regions, ranging from primary sensory areas (visual cortex, Saleem et al., Nature, 2018) to prefrontal areas (Mashhoori et al., *eLife*, 2018). The results presented here linking DDi spiking to active sensing and landmark detection stand by themselves, and this interpretation is not required for the manuscript. In general, the manuscript will benefit from rewriting to tone down statements in several places and also to acknowledge previous literature where appropriate. These specific suggestions are listed below:

The authors should acknowledge that other groups have reported recordings from forebrain neurons in goldfish and found results similar to their own (Representation of Border, Velocity and Speed in the Goldfish Brain. Ehud Vinepinsky, Shay Perchik, Ohad Ben-Shahar, Opher Donchin, Ronen Segev. doi: https://doi.org/10.1101/291013).

Abstract: The first sentence is not general and sets up the false expectation that this manuscript deals with the formation of spatial maps. Active sensing should also be defined as it comes out of the blue. The "novelty of the results" section (i.e., we are the first…) should be toned down based on the comments above. This does not detract from the novelty of the study and helps put it into better context. Also, the last sentence is vague and should be made more precise (i.e., what is meant by "connects active sensing via sensory sampling rate and direct movements…").

Introduction: please tone down the statement towards the end of the second paragraph and instead connect with existing literature while emphasizing some of the unique advantages of weakly electric fish (i.e., their EOD). It should be made clear at the onset that the authors are studying naive fish (this is first mentioned in the discussion and comes too late), this needs to be better connected with the Jun et al., 2016 results.

Results: Please reword vague statements like "…are likely linked to both sensory and motor activity" (end of the first paragraph) to make them more precise. Please include a preamble explaining what was done (esp. that you are using naive fish), and describe the environment in more details. Also explain how you removed landmarks (i.e., in the animal's presence of absence).

Discussion: Overall, it is assumed throughout that results obtained in Apteronotus will apply to Gymnotus and vice-versa. This needs to be stated and justified. In particular, the results of Wallach et al., 2018, were obtained in immobilized Apteronotus and it is not clear whether these cells would respond similarly during movement. The results also need to be properly discussed with regard to those of Vinepinsky et al., 2018. Finally, the authors suggest that DDi is most like area CA3 of the Hippocampus. Can they elaborate on that and provide more links between their results and the vast literature on this area? Further, the discussion about the flow of information between PG, DL, DDmg and DDi (Discussion section) could benefit from a proposed circuit model diagram. The Discussion could also include a few lines about recording of neural activity in freely navigating animals and fish in particular.

Materials and methods: The description of methods especially for Figures 4B and 7 are hard to follow. For example, it is not clear in Figure 4B, how the gray distribution was derived, if there are no peaks in the black trace of Figure 4A. Similarly for Figure 7 in the section "Swimming direction and landmark location preference analysis" especially for STLM and PTLM can be made more accessible perhaps by making a schematic for the data transformation.

---

## [Author Response]

Essential revisions:The initial part of the manuscript shows that DDi spikes are sparse, occur only during swimming and active sensing, and are preceded by increased EOD sampling density (Figures 3 and 4). The later part of the manuscript presents data to argue that DDi cells detect landmarks and swim directional and positional preferences. But these two aspects are not sufficiently interwoven to generate an understanding of the role of DDi neurons during active sensing.

We have now added a paragraph to the Results section (second paragraph) that links the different sections in and provides the logical flow between them. We hope that this interweaves the different aspects of our paper together sufficiently.

For neurons that have significant stEODr and landmark specificity, what exactly would these neurons be coding for?

This is a very important question. We are not certain what exactly these neurons are coding for yet. DDi does not directly receive electrosensory motion signals and does not directly project to the motor output (tectum) region. It receives input from DL and then projects back to DL indirectly, via DDmg. We believe that the DDi to DL feedback is important for the storage of spatial memories in DL (and we mention this in the Discussion) but we simply don’t know enough about DL to have convincing hypotheses as to what DDi neurons are coding for. That said, based on the following observations, we suspect that they encode high-order information about the structure of the environment pertaining to spatial navigation.

First, these units were active only when the fish was swimming around the tank and otherwise silent (Figure 4).

Second, most of the spikes we recorded occurred in the vicinity of landmarks i.e. when fish was less than 10 cm away from landmarks (Figure 6—figure supplement 1).

Third, spatial firing structure and the amount of place information conveyed by the units often varied with changing land mark configuration (Figure 7), e.g. the firing rate often decreased around the location of a removed landmark. Figure 7—figure supplement 1 additionally highlights the effect of adding landmarks to the tank at locations away from a previously present landmark, i.e. the home structure, on the firing rate of the neurons around this structure. Adding landmarks resulted in a decrease in firing rate around the location of home (despite the home still being present) in some units (significant in the third unit in Fish 2 and first unit in Fish 4, some other units showed a similar trend but not significant). In some units the rate increased eventually over few hours of landmark stability (each trial was ~1 hour long), whereas in others it remained low. Most units showed a reduction in the firing rate around the home region again once the home was removed. Therefore, the firing rate of some neurons around a given landmark can be modulated not only by the removal of the landmark, but also as a result of changes in other parts of the environment.

In summary, spiking of neurons in DDi is strongly linked to swimming around landmarks and is globally modulated by the change in landmarks structure. We therefore hypothesize that spiking in these neurons is not a simple electrosensory representation of objects although what exactly they encode remains to be discovered.

We think that it would not be appropriate to add these vague ideas and the rather extensive discussion above and would prefer not to speculate so far beyond our data. We have added a few sentences along the lines above and have added references to the new supplementary figures in the Results section (subsection “Spiking occurred near boundaries and landmarks”).

More quantitative analyses looking at causality as well as interactions between EOD rate and other variables (position, swim speed, etc.) are needed to better assess what is causing spiking in these neurons.

There was no causal relationship between increases in EODr and spiking of DDi neurons. Although on average DDi spikes were linked to a peak in EODr this was not true for every single spike. As shown in Figure 5B, the peaked histograms of the timing of EODr, Speed and Smpdens peaks relative to individual spikes, had a relatively large width, indicating spike-to-spike variability. This finding is perhaps not surprising as there is no reason to believe that fluctuations in EODr that occur at a much higher rate than spiking rate of DDi neurons should causally relate to individual spikes we recorded from the small group of DDi cells. We have added this information to the Results section (subsection “Sampling density”, last paragraph).

We next asked whether spikes that occurred in close vicinity of landmarks (<3 cm from landmarks or tank boundary, which corresponds to object’s reliable electrosensory ‘detectability’ range, Jun et al., 2016) were on average better linked to peaks in EODr than those that occurred far away from them (>10 cm away). We found that for all but three units (1 unit in fish 3 and two units in fish 5) spike-triggered EOD rates were higher when the fish was very close to landmarks or tank boundary (Figure 5—figure supplement 2). Both fish 3 and 5 spent most of their time near the tank boundary and therefore only a few spikes were fired far from landmarks/boundary. Interestingly, for most units the peak in the average stEODr was not absent when the fish was far from all landmarks. The coupling between DDi spikes and EODr transients could therefore occur at locations away from landmarks, albeit less strongly. We have added this information to the Results section (subsection “EODr”, last paragraph).

Further, they should test whether different variables (e.g., swim speed and EOD rate) are correlated, as this would then trivially explain why some neurons respond to changes in both variables.

This is a very good point and we found that EOD rate and swimming speed were indeed significantly correlated although the strength of this correlation was variable across animals (Figure 5—figure supplement 3). Moreover, the relationship between the two could not be fully explained with a linear function. In three out of 5 fish, less than ~13% of the variance in EODr could be explained by the variance in speed. It is important to note that the fish that showed the highest level of correlation and R^2^ (Fish 5) possessed the least sharp peaks in stEODr and stSpeed (Figure 5—figure supplement 1, Fish 5). And fish with lower correlation levels (e.g. Fish 2 and 4) possessed sharper peaks in the spike triggered averages (Figure 5—figure supplement 1, Fish 2 and 4). We therefore conclude that although EODr and Speed could be correlated in some instances, this correlation is not enough to explain the relationship between these two variables and spiking of DDi neurons. This analysis is presented in Figure 5—figure supplement 3 and mentioned in the Results section (subsection “Swim speed”).

The statistics used assume stationary data. However, this is unlikely to be the case for the dataset considered, as these are naive fish that are discovering a new environment. The authors need to assess nonstationarities in their data and test how this will affect their results. If there are nonstationarities, the random-shifted spike trains are likely to have different statistics than the original one.

The average firing rate of some neurons were indeed significantly affected by changes in landmark configuration as evident by significant changes in firing rate of neurons across trials as pointed out in Figure 6 in the context of landmark removal (Also see Figure 7—figure supplement 1). The firing rate within a trial, however, did not change significantly for most recorded units (Figure 6—figure supplement 3, except for the second trial in the second unit in fish 2 and first trial in the first two units and the third trial in the third unit in fish 4). Therefore, although firing of neurons across trials could have been non-stationary, this was not the case within a trial for most trials. We have briefly summarized our conclusions with respect to where the data was stationary and where it was not in the Materials and methods section (subsection “Analysis of fish’s swimming trajectory and spatial firing rates”, second paragraph).

All statistics for assessing significance of place-specificity were performed within trials by comparing place information calculated for a given spike train versus that calculated for randomly time shifted versions of that spike train keeping the relative spike timings the same. It is true, that due to small number of spikes, if the firing rate of a unit was higher at a certain time period (this could be e.g. due to visiting a certain location only at that time), randomly time-shifting the whole spike train would preserve this bias and yield high information levels even for the time-shifted data. Using this method for testing the significance of place information therefore, provides a conservative estimate for the p values. Indeed, when we recalculated the significance of place information, this time using randomly shuffled spike times (as opposed to randomly time shifting the whole train) we found that more units showed a significance in their place information (11 as opposed to 8. see also response to the last minor point). We have added this information to the Materials and methods section (subsection “Analysis of fish’s swimming trajectory and spatial firing rates”, third paragraph).

In the analysis of spike-triggered EODr, speed and Sampling density random shifts were similarly performed within trials and then stEODr, stSpeed and stSmpdens were averaged across all trials and spikes of each unit. Across-trials non-stationarities in the firing rate may indeed affect the averages calculated in this manner. We repeated this analysis calculating the averages only within trials. Figure 5—figure supplement 4 shows these averages for units that showed significant stEODr peaks. The left column shows the average when calculated pooling across all trials and the other column corresponds to individual trials. For many trials the shapes of stEODr, stSpeed and stSmpdens for a given unit were like what we found when we averaged across trials (Figure 5—figure supplement 4, compare columns on the right to the left-most column). For some other trials the peaks in these measures were less prominent. This result is another manifestation of the fact that there was no 1-1 causal relation between a peak in EODr and spiking of DDi neurons and that their relative timing was variable as indicated by the histogram of in Figure 5B. There was no relation between presence or absence of a clear peak in trial-by-trial calculated spike-triggered averages and whether changes were induced in landmark configuration on that trial. The average stEODr, stSpeed and stSmpdens for randomly time-shifted data were similar when pooled across all trials or when calculated within a trial (Figure 5—figure supplement 4, black traces), and they were similarly flat when calculated for randomly shuffled spikes (data not shown). This information has been added to the Materials and methods section (subsection “Analysis of EOD rate – spike relationship”.

The reviewers felt that the evidence that this is a place-like response related to spatial navigation based on path integration is quite weak. However, evidence presented in the manuscript convincingly shows that DDi spiking is related to an active sensing based detection mechanism, and encodes landmark locations based on EOD and B-scans. The authors allude to path integration in the Introduction and Discussion, and draw a comparison between DDi firing and hippocampal place cell responses. Spatially specific firing related to landmarks is not sufficient evidence for path-integration based navigation, and indeed, spatially specific firing is seen in many mammalian brain regions, ranging from primary sensory areas (visual cortex, Saleem et al., Nature, 2018) to prefrontal areas (Mashhoori et al., eLife, 2018). The results presented here linking DDi spiking to active sensing and landmark detection stand by themselves, and this interpretation is not required for the manuscript. In general, the manuscript will benefit from rewriting to tone down statements in several places and also to acknowledge previous literature where appropriate. These specific suggestions are listed below:

We agree with the reviewers that we do not provide evidence in this paper for role of DDi neurons in path-integration. We have removed the reference to path integration from the manuscript and now focused on the relation between spiking and active sensing measures. Further, we have now completely re-written the Introduction and Discussion to focus on the connection of our results to active sensing.

The authors should acknowledge that other groups have reported recordings from forebrain neurons in goldfish and found results similar to their own (Representation of Border, Velocity and Speed in the Goldfish Brain. Ehud Vinepinsky, Shay Perchik, Ohad Ben-Shahar, Opher Donchin, Ronen Segev. doi: https://doi.org/10.1101/291013).

Thanks for bringing this to our attention, we have added this reference to the Discussion section (eighth paragraph).

Abstract: The first sentence is not general and sets up the false expectation that this manuscript deals with the formation of spatial maps. Active sensing should also be defined as it comes out of the blue. The "novelty of the results" section (i.e., we are the first…) should be toned down based on the comments above. This does not detract from the novelty of the study and helps put it into better context. Also, the last sentence is vague and should be made more precise (i.e., what is meant by "connects active sensing via sensory sampling rate and direct movements…").

We agree with the reviewer. We have completely re-written the Abstract and have now added explanation about active sensing to and removed the sentences pertaining to formation of spatial maps. As per the novelty of the results, this is certainly the first report on recordings from the CA3-like region of the pallium (DD) as recordings in other studies were obtained from a different pallial region (DL). We have changed the Abstract to point this out more clearly.

Introduction: please tone down the statement towards the end of the second paragraph and instead connect with existing literature while emphasizing some of the unique advantages of weakly electric fish (i.e., their EOD). It should be made clear at the onset that the authors are studying naive fish (this is first mentioned in the discussion and comes too late), this needs to be better connected with the Jun et al., 2016 results.

We have rewritten the Introduction to better emphasise the advantages of electric fish and added information about the naivety of the fish in the fourth and last paragraphs of the Introduction. The Introduction was completely re-written.

Results: Please reword vague statements like "…are likely linked to both sensory and motor activity" (end of the first paragraph) to make them more precise. Please include a preamble explaining what was done (esp. that you are using naive fish), and describe the environment in more details. Also explain how you removed landmarks (i.e., in the animal's presence of absence).

We have reworded that statement (Results subsection “Spikes were sparse and occurred mainly during movement”) and added more information about experimental setup and manipulations to the Materials and methods subsection “Experimental setup and trials”.

Discussion: Overall, it is assumed throughout that results obtained in Apteronotus will apply to Gymnotus and vice-versa. This needs to be stated and justified. In particular, the results of Wallach et al., 2018, were obtained in immobilized Apteronotus and it is not clear whether these cells would respond similarly during movement. The results also need to be properly discussed with regard to those of Vinepinsky et al., 2018. Finally, the authors suggest that DDi is most like area CA3 of the Hippocampus. Can they elaborate on that and provide more links between their results and the vast literature on this area? Further, the discussion about the flow of information between PG, DL, DDmg and DDi (Discussion section) could benefit from a proposed circuit model diagram. The Discussion could also include a few lines about recording of neural activity in freely navigating animals and fish in particular.

We have added discussion about similarities between Apteronotus and Gymnotus species to the Discussion (third paragraph). We have also discussed our results with respect to those of Vinepinsky et al., 2018, in Discussion (eighth paragraph). We have added a new figure (Figure 1) that shows this circuit diagram and illustrates the pattern of connectivity/information flow among electrosensory regions. The statement that DDi is homologous to CA3 region is based on Elliot et al., 2017; we have summarized the essential conclusion of this study in the Introduction (fourth paragraph) as well as in the new Figure 1 legend. We have also added appropriate references to Elliott et al. throughout the discussion. We have also added a few lines on the importance of recordings of pallial activity in freely moving animals while monitoring active sensing in the Discussion section (last paragraph).

Materials and methods: The description of methods especially for Figures 4B and 7 are hard to follow. For example, it is not clear in Figure 4B, how the gray distribution was derived, if there are no peaks in the black trace of Figure 4A. Similarly for Figure 7 in the section "Swimming direction and landmark location preference analysis" especially for STLM and PTLM can be made more accessible perhaps by making a schematic for the data transformation.

We apologize for these points not being clear. The black trace in Figure 4A (Now Figure 5A) is the average of EODr traces in a window around individual spikes from the randomly time-shifted spike trains. The timing of EODr peaks were randomly distributed relative to the randomized spike times and therefore, the average did not show a peak. The gray distribution therefore does not show a bias in the probability of peak times. We have now added further explanation for methods used for generating the histogram in the Materials and methods section (subsection “Analysis of EOD rate – spike relationship”. We have also added a panel to the new Figure 8 (previously Figure 7) to illustrate the procedure for generating STLM and PTLM and added further details on procedure within the figure legend and in the Materials and methods section (subsection “Swimming direction and landmark location preference analysis”.